# Wnt signaling enhances macrophage responses to IL-4 and promotes resolution of atherosclerosis

Ada Weinstock[1], Karishma Rahman[1], Or Yaacov[1], Hitoo Nishi[1], Prashanthi Menon[1], Cyrus A Nikain[1], Michela L Garabedian[1], Stephanie Pena[1], Naveed Akbar[2], Brian E Sansbury[3], Sean P Heffron[1,4], Jianhua Liu[5], Gregory Marecki[1], Dawn Fernandez[6], Emily J Brown[1], Kelly V Ruggles[7], Stephen A Ramsey[8], Chiara Giannarelli[6,9,10], Matthew Spite[3], Robin P Choudhury[2,11], P'ng Loke[12,13†], Edward A Fisher[1,4,14]*

[1]Department of Medicine, Leon H. Charney Division of Cardiology, Cardiovascular Research Program, New York University Grossman School of Medicine, New York, United States; [2]Division of Cardiovascular Medicine, Radcliffe Department of Medicine, University of Oxford, Oxford, United Kingdom; [3]Center for Experimental Therapeutics and Reperfusion Injury, Department of Anesthesiology, Perioperative and Pain Medicine, Brigham and Women's Hospital and Harvard Medical School, Boston, United States; [4]NYU Center for the Prevention of Cardiovascular Disease, New York University Grossman School of Medicine, New York, United States; [5]Department of Surgery, Mount Sinai School of Medicine, New York, United States; [6]Cardiovascular Research Center, Department of Medicine, Icahn School of Medicine at Mount Sinai, New York, United States; [7]Division of Translational Medicine, Department of Medicine, New York University Langone Health, Institute for Systems Genetics, New York University Grossman School of Medicine, New York, United States; [8]Department of Biomedical Sciences, School of Electrical Engineering and Computer Science, Oregon State University, Corvallis, United States; [9]The Precision Immunology Institute, Icahn School of Medicine at Mount Sinai, New York, United States; [10]Department of Microbiology (Parasitology), New York University School of Medicine, New York, United States; [11]Department of Genetics and Genomic Sciences, Icahn School of Medicine at Mount Sinai, New York, United States; [12]Acute Vascular Imaging Centre, Radcliffe Department of Medicine, University of Oxford, Oxford, United Kingdom; [13]Laboratory of Parasitic Diseases, National Institute of Allergy and Infectious Diseases, National Institutes of Health, Bethesda, United States; [14]Departments of Cell Biology and Microbiology, New York University Grossman School of Medicine, New York, United States

*For correspondence:
Edward.Fisher@nyulangone.org

Present address: † Laboratory of Parasitic Diseases, National Institute of Allergy and Infectious Diseases, National Institutes of Health, Bethesda, United States

Competing interests: The authors declare that no competing interests exist.

**Abstract** Atherosclerosis is a disease of chronic inflammation. We investigated the roles of the cytokines IL-4 and IL-13, the classical activators of STAT6, in the resolution of atherosclerosis inflammation. Using $Il4^{-/-}Il13^{-/-}$ mice, resolution was impaired, and in control mice, in both progressing and resolving plaques, levels of IL-4 were stably low and IL-13 was undetectable. This suggested that IL-4 is required for atherosclerosis resolution, but collaborates with other factors. We had observed increased Wnt signaling in macrophages in resolving plaques, and human genetic data from others showed that a loss-of-function Wnt mutation was associated with premature atherosclerosis. We now find an inverse association between activation of Wnt signaling and disease severity in mice and humans. Wnt enhanced the expression of inflammation resolving

factors after treatment with plaque-relevant low concentrations of IL-4. Mechanistically, activation of the Wnt pathway following lipid lowering potentiates IL-4 responsiveness in macrophages via a PGE$_2$/STAT3 axis.

## Introduction

The past several decades have seen large reductions in fatal and non-fatal complications arising from atherosclerosis, but the most recent data show that these are on the rise again (*Xu et al., 2016*). According to the World Health Organization, atherosclerotic cardiovascular disease is still the leading cause of morbidity and mortality worldwide (*WHO, 2017*). Current therapies aim at inhibiting further progression of the disease, most commonly by lowering low-density lipoprotein (LDL) cholesterol levels. When most patients present for cardiovascular risk reduction, they already have advanced, complex and potentially dangerous plaques. Indeed, several clinical trials examining the effectiveness of available therapies concluded that the majority of treated patients will still undergo major coronary events (*Libby, 2005*). Hence, the resolution of atherosclerosis remains a vital clinical goal.

We and our collaborators have previously demonstrated a range of resolution of atherosclerosis in several mouse models (*Rahman et al., 2017*; *Peled et al., 2017*; *Feig et al., 2011a*; *Chereshnev et al., 2003*; *Rahman and Fisher, 2018*; *Potteaux et al., 2011*; *Basu et al., 2018*). We have utilized these models to identify and investigate the involved pathways. Collectively, our studies demonstrated that resolution entails a net decrease in plaque macrophage content, with the remaining macrophages showing a pro-resolving phenotype (*Rahman and Fisher, 2018*). For example, during disease progression, plaque macrophages are predominantly inflammatory, expressing factors such as C-C motif chemokine ligand 2 (CCL2) 1, tumor necrosis factor (TNF) α, and vascular cell adhesion molecule (VCAM) 1 (*Parathath et al., 2011*; *Cybulsky et al., 2001*; *Combadière et al., 2008*; *Boyle et al., 2003*; *Feig et al., 2011b*). Conversely, during resolution there is an accumulation of inflammation-resolving, tissue remodeling macrophages, which express arginase (ARG) 1 and mannose receptor (MRC) 1, and the concomitant loss of macrophages with the inflammatory subtype (*Feig et al., 2011b*; *Hewing et al., 2013*).

ARG1 has been previously considered to be a beneficial factor in atherosclerosis. A potential basis for this is that increased ARG1 activity would limit substrate to iNOS for conversion to damaging reactive nitrogen species (*Munder, 2009*). More recently, ARG1 was shown to aid the licensing of macrophages to clear multiple apoptotic cells, which would be expected to be important in atherosclerosis resolution (*Yurdagul et al., 2020*). *Arg1* expression is mainly regulated through the canonical transcription factor for type 2 immune responses, signal transducer and activator of transcription (STAT) 6 (*Van Dyken and Locksley, 2013*; *Rutschman et al., 2001*). Importantly, we recently showed that STAT6 is necessary for resolution of murine atherosclerosis in a process that also requires monocytes to be recruited to resolving plaques (*Rahman et al., 2017*).

In the present study, we aimed at determining the signals upstream of STAT6 that induce resolution of atherosclerosis and hypothesized that the classical activators of STAT6, IL-4, and/or IL-13 are involved. Our data show that IL-4 is indeed necessary, but that there are also additional factors operating in concert with it, one of which is Wnt.

## Results

### Deficiency in *Il4/13* impairs the resolution of atherosclerosis

To begin to understand the factors upstream of STAT6 needed for atherosclerosis resolution, we examined the requirements for the canonical activators of STAT6, namely IL-4 and IL-13. Mice deficient in both *Il4* and *Il13* (*Il4$^{-/-}$Il13$^{-/-}$*) were injected with a liver-specific AAV expressing *Pcsk9* to result in LDLr deficiency and fed Western diet (WD) to promote hypercholesterolemia in order to develop advanced atherosclerotic lesions by 20 weeks (*Bjørklund et al., 2014*; *Roche-Molina et al., 2015*; *Peled et al., 2017*). At this time, a baseline (BL) group was harvested to examine the size and composition of the established plaques. To decrease plasma cholesterol thereafter and induce resolution of atherosclerosis (Res), mice were injected with an anti-sense oligonucleotide (ASO) to

apolipoproteinB (ApoB) and switched to chow diet for 3 weeks, as described (*Lin et al., 2019*; *Sharma et al., 2020*; *Figure 1A*).

Treatment with the *Pcsk9* AAV and WD caused a significant increase in total plasma cholesterol (~500 mg/dL), whereas subsequent ApoB ASO injections decreased plasma cholesterol to normal levels (~60 mg/dL, *Figure 1B*). We previously showed, in an identical experimental design, that ApoB ASO administration to *Pcsk9* AAV and WD-treated WT mice for 3 weeks resulted in decreased plaque size and macrophage (CD68+ cells) content in aortic roots (*Sharma et al., 2020*). In the present study, following cholesterol lowering, there were no significant changes in aortic root plaque size (*Figure 1C*), macrophage content (*Figure 1D–F*), or necrotic core area (*Figure 1G*) in mice deficient in IL-4 and IL-13. Collectively, these data indicate that IL-4/13 are needed for atherosclerosis resolution.

## IL-4/13 produced by cells newly recruited to plaques upon cholesterol lowering are dispensable for resolution of atherosclerosis

We previously reported that atherosclerosis resolution requires recruitment of new *Stat6*-expressing monocytes to plaques (*Rahman et al., 2017*). Because we demonstrated that IL-4/13 are important for atherosclerosis resolution (*Figure 1*), we next examined whether the source of the IL-4/13 is from cells newly recruited in the resolution phase or from cells in the plaques prior to the induction of lipid lowering.

Thus, we performed aortic arch transplant studies (as described previously *Rahman et al., 2017*; *Chereshnev et al., 2003*), in which aortic arches from hyperlipidemic *Ldlr*[-/-] donors fed WD for 20 weeks (BL group) were transplanted into normolipidemic WT or *Il4*[-/-]*Il13*[-/-] mice. Four weeks post transplant, the grafted aortic segments were evaluated for plaque burden and macrophage content (see Experimental design; *Figure 2A*). As expected, the arch donors (and BL group) had elevated total cholesterol levels (978 ± 141 mg/dL), whereas the transplant recipients were normocholesterolemic (79 ± 22 mg/dL and 73 ± 10 mg/dL for WT and *Il4*[-/-]*Il13*[-/-] mice, respectively; *Figure 2B*). Although plaque size did not change significantly between the BL and transplant recipients (*Figure 2C*), there was a significant and comparable reduction (~60%) in plaque macrophages in both the WT and *Il4*[-/-]*Il13*[-/-] recipient groups (*Figure 2D, E*). The necrotic core area significantly diminished as well in both recipient groups (*Figure 2F*). Thus, resolution of atherosclerosis was similar in WT and *Il4*[-/-]*Il13*[-/-] recipients (*Figure 2D–G*), indicating that IL-4/13 production by cells newly recruited is not essential for resolution of atherosclerosis. These results were confirmed in an additional aortic transplant experiment, in which plaque-bearing aortic arches from *Apoe*[-/-] donors fed WD for 16 weeks were grafted into either WT or *Il4*[-/-] recipients for 2 weeks (*Figure 2—figure supplement 1A*), resulting in plaque size and macrophage content changes consistent with those in *Il4*[-/-]*Il13*[-/-] transplant recipients (*Figure 2—figure supplement 1B–G*). Notably, there was a marked decrease in macrophage content of both WT and *Il4*[-/-] arch recipients (*Figure 2—figure supplement 1D–E, G*).

Our data thus far indicate that IL-4/13 are needed for atherosclerosis resolution, and the source of these cytokines is not from cells that are newly recruited in the resolution phase, but from cells accumulating during the disease progression phase.

## IL-4/13 content in plaques is unchanged between atherosclerosis progression and resolution

We next evaluated whether IL-4/13 content is increased during atherosclerosis resolution. To measure cytokine abundances in plaques, *Ldlr*[-/-] mice fed WD for 20 weeks (BL) and treated with ApoB ASO for an additional 3 weeks (Res) were examined. Upon harvest, plaques from aortic arches and brachiocephalic arteries were carefully excised and evaluated for CCL2, IL-4, and IL-13. CCL2 was used as a control since we previously showed that its levels are decreased in resolving plaques (*Rahman et al., 2017*; *Feig et al., 2011b*). Consistent with our previous reports (e.g., *Rong et al., 2001*), CCL2 abundance in plaques showed a marked decrease from progression to resolution (*Figure 3A*).

In contrast to CCL2, IL-4 levels did not significantly change between progression (111 pg/mL) and resolution (123 pg/mL; *Figure 3A*) and were strikingly lower (over two orders of magnitude) than those commonly used in cell culture experiments (10–20 ng/mL) (*Minutti et al., 2017*;

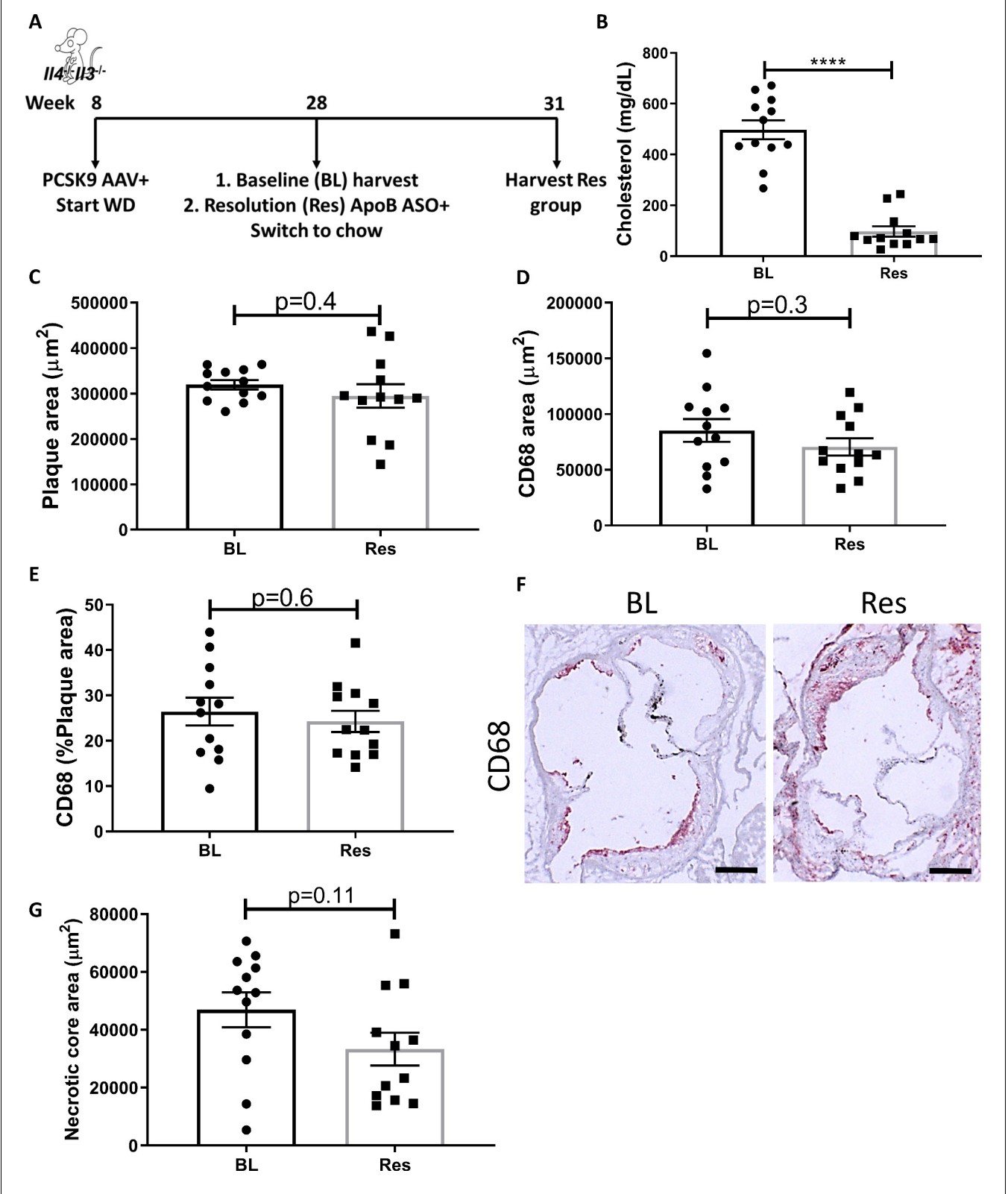

**Figure 1.** Global IL-4/13 deficiency impairs atherosclerosis resolution. (**A**) Experimental design. (**B**) Plasma cholesterol. Aortic root morphometric analysis for areas of (**C**) plaque, (**D**) CD68, (**E**) % of plaque positive for CD68, (**F**) representative images (red, CD68; scale bar, 200 µm), and (**G**) necrotic core. ****p<0.0001, as determined via two-tailed Student's *t*-test.

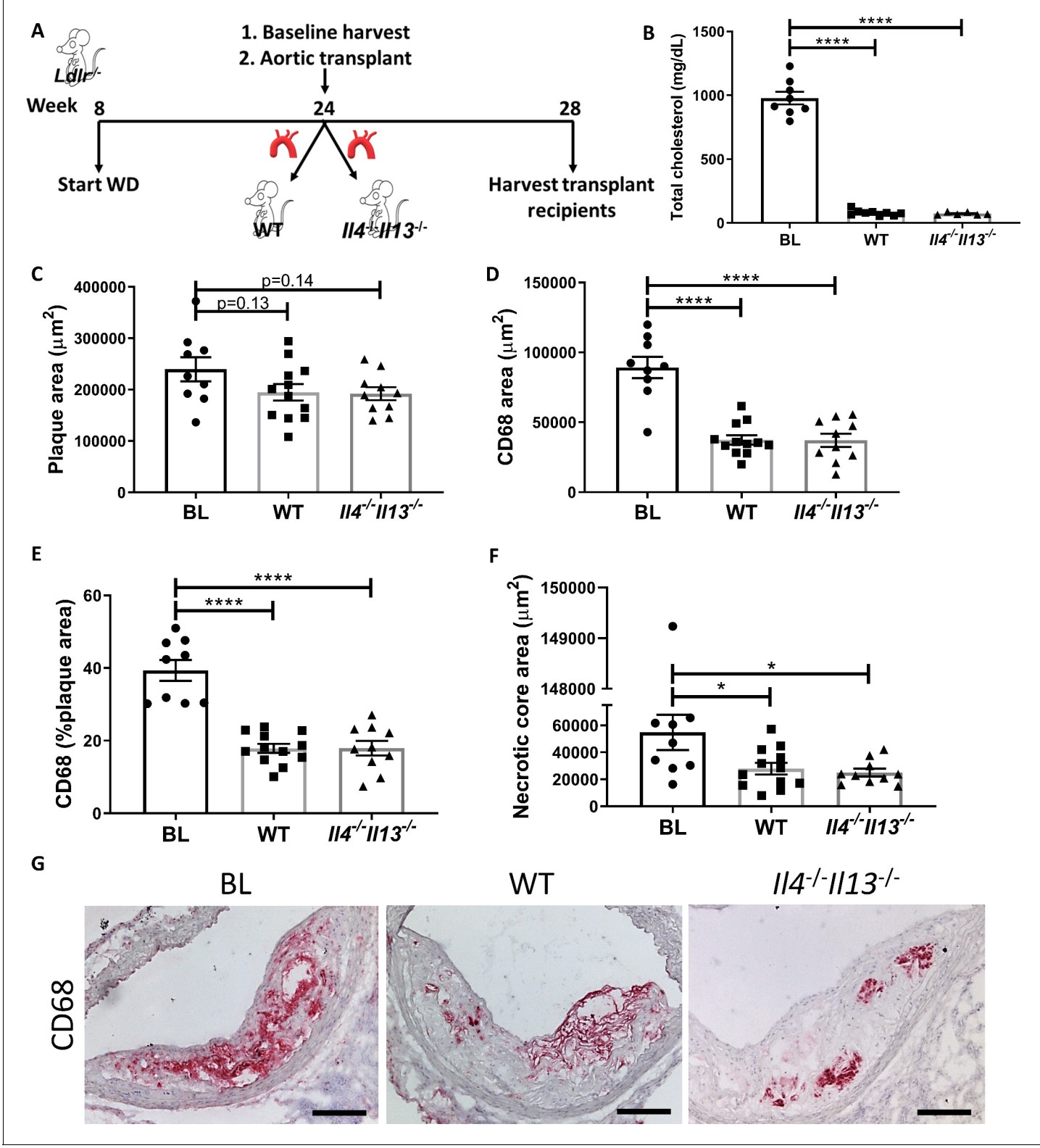

**Figure 2.** Production of IL-4/13 by newly recruited cells is not required for resolution of atherosclerosis. (**A**) Experimental design. (**B**) Plasma cholesterol. (**C–G**) Morphometric analysis and representative images of grafted aortic arches for areas of (**C**) plaque, (**D**) CD68, (**E**) % of plaque positive for CD68, and (**F**) necrotic core. Red, CD68; scale bar, 100 μm. ****p<0.0001 and *p<0.05 compared with the baseline group as determined via one-way ANOVA and Dunnett's multiple comparison test.

*Figure 2 continued on next page*

**Figure supplement 1.** Production of IL-4 by newly recruited cells is not required for resolution of atherosclerosis.

*Bosurgi et al., 2017*; *Sanin et al., 2018*). Interestingly, IL-13 was undetectable (<3 pg/mL) in both progressing and resolving plaques. To confirm that plaque IL-13 levels are negligible, we examined its expression in a recently published dataset of plaque immune cells pre- and post-atherosclerosis resolution (*Sharma et al., 2020*). In accordance with the undetectable protein levels, only one cell in

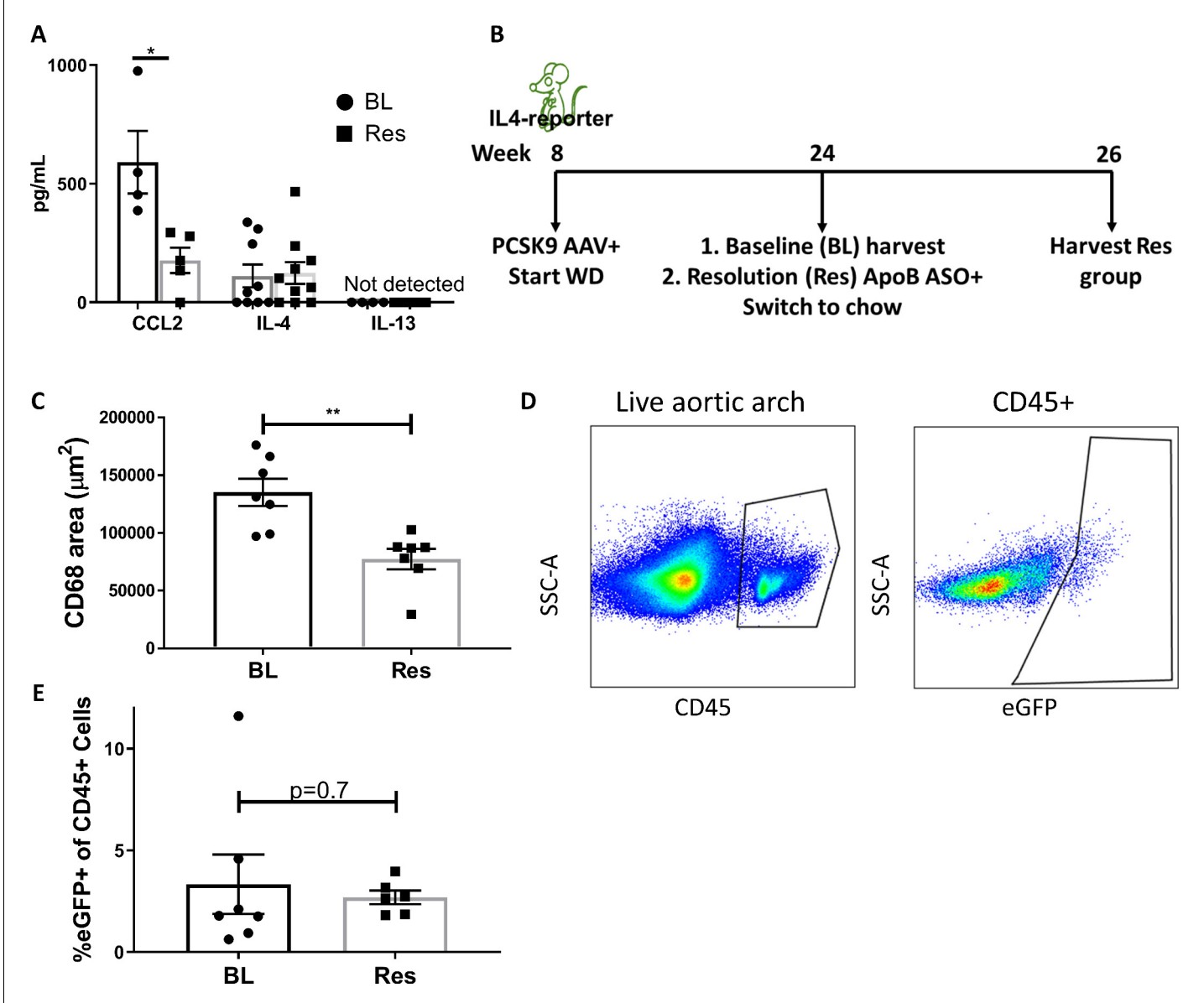

**Figure 3.** IL-4/13 levels do not change during resolution of atherosclerosis. (A) Quantification of CCL2, IL-4, and IL-13 protein levels in aortic arch plaques of *Ldlr⁻/⁻* BL mice or further treated for lipid lowering for 3 weeks (Res). (B) Experimental design. (C) CD68-positive area in aortic roots. (D) Gating scheme of aortic arches single-cell suspensions and (E) quantification of IL-4 (eGFP+) producing cells in aortic arches. **p<0.01 and *p<0.05 determined by an unpaired *t*-test (two-tailed).

**Figure supplement 1.** *Il13* expression in plaques during disease progression and resolution on a single cell level.

the entire dataset showed *Il13* expression (*Figure 3—figure supplement 1*). Hence, we focused on IL-4 in subsequent experiments.

Because IL-4 production is needed in cells that accumulate during atherosclerosis progression, but not in cells newly recruited during resolution (*Figures 1* and *2*), we next examined the abundance of IL-4-producing cells in plaques. IL-4-reporter mice were injected with *Pcsk9*-AAV and fed WD for 16 weeks to promote atherosclerosis formation. BL mice were harvested, and the disease resolution group was induced with ApoB ASO treatment and chow diet (*Figure 3B*). We confirmed that ApoB ASO treatment led to reduction in plaque macrophages and promoted resolution of atherosclerosis (*Figure 3C*). Aortic arches were digested and analyzed using flow cytometry (*Figure 3D*). Results showed no significant difference in the proportion of IL-4-producing leukocytes (CD45+ cells) when comparing mice from the BL (3.3%) and resolution (2.7%) groups (*Figure 3E*).

Our data suggest that small amounts of IL-4-producing cells accumulate in progressing plaques and do not increase further during the resolution phase. In the reparative environment, the IL-4 that accumulated is apparently sufficient to activate STAT6, which in turn polarizes plaque macrophages to pro-resolving cells.

## Macrophage Wnt signaling is associated with disease resolution and decreased atherosclerosis severity in mice and humans

Because IL-4 levels do not change in resolving plaques, we postulated that this environment contains other factors that license IL-4 activation of STAT6.

To explore this, we reviewed our transcriptome data from macrophages taken by laser-capture microdissection from progressing and resolving plaques (*Ramsey et al., 2014*). These data (from two independent mouse models (*Feig et al., 2011a*) suggested activation of the Wnt pathway (*Ramsey et al., 2014*). This was of particular interest since a mutation causing downregulation of this pathway led to early coronary artery disease in humans (*Mani et al., 2007*). Moreover, activation of Wnt signaling in macrophages was recently shown to inhibit atherosclerosis progression (*Wang et al., 2018*).

To confirm this activation in resolving plaques, single-cell RNA-sequencing (scRNA-seq) data from progressing and resolving plaque macrophages that we recently published (*Lin et al., 2019*) were analyzed for the expression of Wnt-related genes (GO term GO0060070 'Wnt signaling Pathway'). The expression of these genes was summed per cell, aggregated by Louvain clusters and plotted (*Figure 4A*). This analysis showed expression of Wnt-related genes in all the macrophage clusters (*Figure 4A*). Notably, comparison between progressing and resolving plaque macrophages demonstrated that in disease resolution there is an overall increase in the expression of Wnt-related genes across most clusters (*Figure 4B*). The relative expression levels of each Wnt-related gene from resolving versus progressing plaque macrophages are presented in *Figure 4C*. Among the Wnt regulated genes, *Cttnb1*, the gene encoding for β-catenin, the downstream canonical effector of Wnt, was the most upregulated across macrophage clusters (*Figure 4C*).

To extend our analysis of Wnt signaling to human plaque macrophages, we analyzed a recently published dataset characterizing the transcriptome of human endarterectomy leukocytes (*Fernandez et al., 2019*). First, Wnt-responsive gene expression data from mouse plaque macrophages (*Figure 4C*) were averaged across all clusters. The 10 highest differentially expressed genes in resolving plaques were identified and their expression examined in carotid plaque macrophages from symptomatic (Sym) versus asymptomatic (Asym) patients (*Fernandez et al., 2019*; *Figure 4D*). Results show that of the 10 genes 7 had lower expression levels in Sym patients' plaque macrophages compared to Asym patients (*Figure 4D*). A permutation test resulted in a p-value=0.008, thereby supporting a *bona fide* inverse relationship between Wnt signaling and disease severity across species, consistent with the genetic loss-of-function association with premature CAD (coronary artery disease; *Mani et al., 2007*).

To independently support the clinical relevance of the inverse relationship between Wnt signaling and disease severity, we next examined whether the Wnt pathway is associated with atherosclerosis resolution in humans. A recently described cohort of statin-naïve patients that were treated with high-dose atorvastatin following presentation with an acute coronary syndrome (*Alkhalil et al., 2018*) was used. Using advanced imaging techniques, the authors evaluated carotid artery plaque volume at presentation and 3 months later. We investigated the expression of seven Wnt-associated genes (*Axin2, Ddx3x, Wls, Senp2, Ctnnb1, Csnk1a1,* and *Amfr*) in peripheral blood

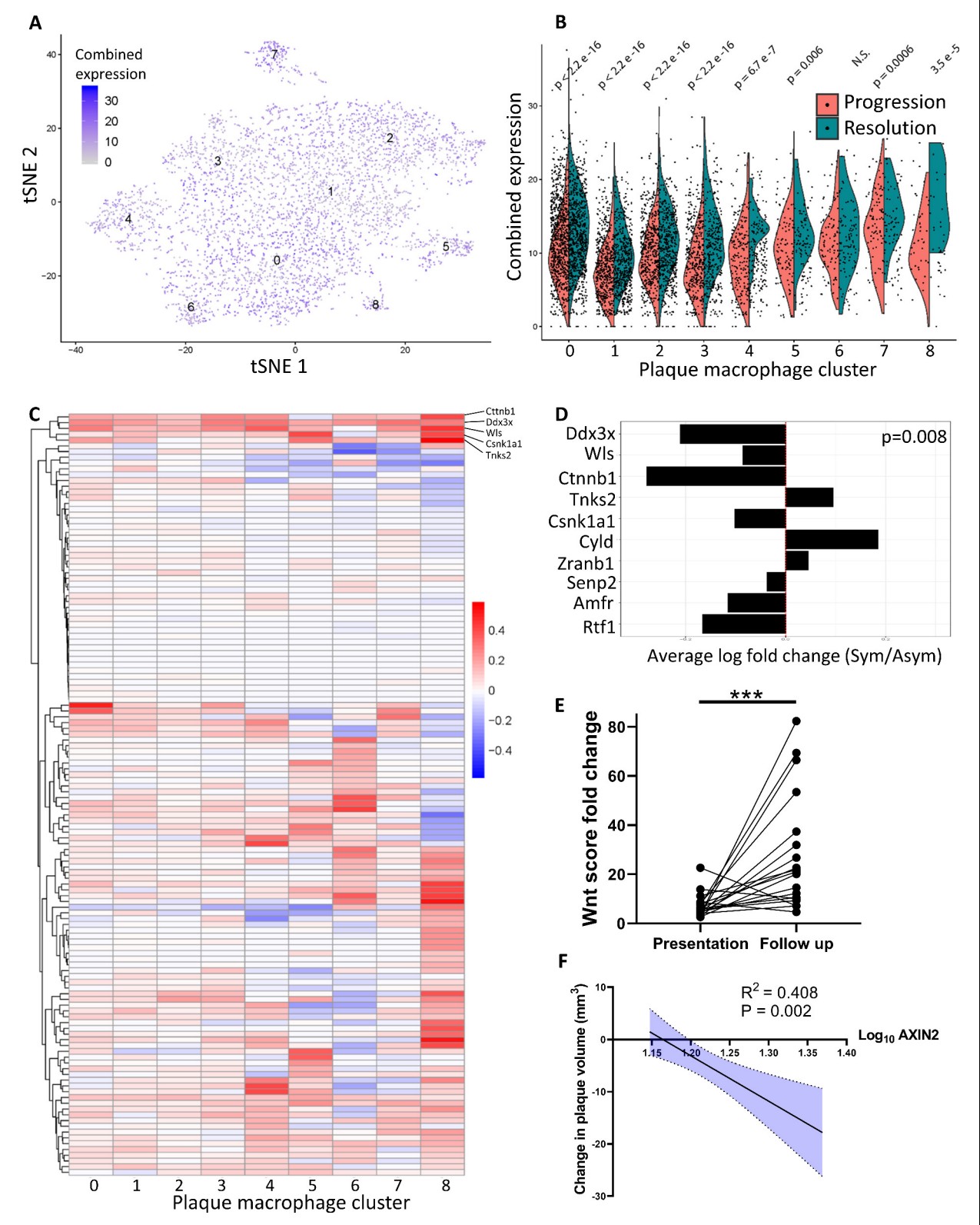

**Figure 4.** Atherosclerosis severity is associated with decreased responsiveness of plaque macrophages to Wnt signaling in mice and humans. (A–C) Cumulative expression of genes associated with the GO term GO:0016055 of data from *Lin et al., 2019* represented in (A) Louvain clusters for both progression and resolution groups and (B) divided by treatment. (C) Individual GO:0016055 gene expression per cluster, comparing treatment groups. Red, higher expression in resolution; blue, higher expression in progression. (D) Human plaque macrophages obtained from endarterectomies (from

*Figure 4 continued on next page*

Figure 4 continued

*Fernandez et al., 2019*) were examined for their expression of the top 10 most upregulated genes from GO:0016055 in mouse resolving plaque macrophages (from *Lin et al., 2019*). Relative expression of symptomatic (Sym) versus asymptomatic (Asym) is presented. Permutation tests of Fernandez et al. expressed genes was performed to calculate the probability of 7 or more genes out of 10 to have lower expression in the Sym group, p=0.008. (E) Expression of seven Wnt marker genes (*Axin2, Ddx3x, Wls, Senp2, Ctnnb1, Csnk1a1,* and *Amfr*) was measured in peripheral blood mononuclear cells (PBMCs) from statin-naïve patients at presentation of acute coronary syndrome and 3 months intensive atorvastatin treatment follow-up (*Alkhalil et al., 2018*). (F) Change in the expression of the Wnt hallmark gene, *Axin2,* in PBMCs obtained from patients was examined in relation to change in plaque volume from presentation to follow-up (*Alkhalil et al., 2018*).

mononuclear cells (PBMCs) obtained at presentation and follow-up. The fold-change of each gene expression at the follow-up compared to presentation was calculated per patient and summed to create a 'Wnt score'. These scores at the 3-month follow-up were significantly increased relative to disease presentation, indicated by the increased Wnt score (*Figure 4E*). Notably, the expression of *Axin2*, the hallmark Wnt target gene (*Jho et al., 2002*; *Lustig et al., 2002*), significantly correlated with a decrease in plaque volume ($R^2$ = 0.408, p=0.002; *Figure 4F*). These data provide further support that atherosclerosis resolution is associated with upregulation of the Wnt pathway not only in mice, but also in humans.

## The Wnt signaling pathway collaborates with IL-4 to enhance expression of macrophage pro-resolving genes in vitro

Because Wnt signaling is upregulated in resolving plaque macrophages and IL-4 levels were similar in progressing and resolving plaques, we examined whether this pathway augments the effects of IL-4. Wnt3a is a wnt pathway activator and is most commonly used for studies in vitro. Thus, we treated bone marrow-derived macrophages (BMDMs) with recombinant Wnt3a protein and IL-4 (at its plaque concentration). First, we confirmed that treatment with Wnt3a activated the Wnt pathway, indicated by the upregulation of *Axin2* (*Figure 5—figure supplement 1A*) and the accumulation of the transcription co-activator β-catenin (*Figure 5—figure supplement 1B*). Examination of genes associated with pro-resolution (e.g., *Arg1*, *Ccl17,* and *Socs1*) revealed that Wnt3a treatment did not alter their expression. However, when cells were co-treated with Wnt3a and IL-4, these transcripts showed increased expression relative to IL-4 alone (*Figure 5A–C*).

To examine whether the enhanced expression of these pro-resolving genes is due to Wnt3a-induced production of IL-4/13, we repeated the in vitro experiments with *Il4*$^{-/-}$*Il13*$^{-/-}$ BMDMs. The data show that the expression of the pro-resolving genes was enhanced in cells co-treated with Wnt3a and IL-4 (*Figure 5D–F*), indicating that the increased gene expression cannot be explained by induced production of IL-4/13. Since the signal transducer of IL-4 is STAT6 (as noted above, previously shown to be required for plaque resolution *Rahman et al., 2017*), we studied its involvement in the enhanced pro-resolving gene expression by using BMDMs from *Stat6*$^{-/-}$ mice. As shown in *Figure 5G–I*, *Stat6* deficiency completely abrogated the expression of *Arg1*, *Ccl17,* and *Socs1* and the enhancement observed in cells co-treated with Wnt3a and IL-4. In sum, the data suggest that Wnt signaling cooperates with IL-4/STAT6 signaling to augment macrophage pro-resolving gene expression.

## Wnt signaling is required for the resolution of atherosclerosis

Our in vitro and in vivo data suggest a role for the Wnt signaling pathway in resolving atherosclerosis. To strengthen this suggestion, we performed an atherosclerosis resolution study. Reversa mice (*Feig et al., 2011a*) were fed WD for 16 weeks to establish advanced atherosclerotic plaques (baseline group; BL). LDL-C levels were then lowered and mice treated with an established Wnt inhibitor, LGK974 (*Liu et al., 2013*), or vehicle, for an additional 3 weeks (*Figure 6A*).

BL mice were hypercholesterolemic (~810 mg/dL), and mice from both lipid-lowering groups were normocholesterolemic (~140 mg/dL; *Figure 6B*). Mice treated with LGK974 had similar body weights (*Figure 6—figure supplement 1A*), lipid profiles (*Figure 6B*), hematological parameters (*Figure 6—figure supplement 1B–E*), and frequencies of Ly6C$^{hi}$ and Ly6C$^{lo}$ circulating monocytes (*Figure 6—figure supplement 1F*) compared to the control (Ctrl) group. To confirm that Wnt inhibition was successful in the LGK974-treated mice, plaques were stained for the active form of β-catenin (*Figure 6—figure supplement 1G*), which showed significantly less active β-catenin. To verify

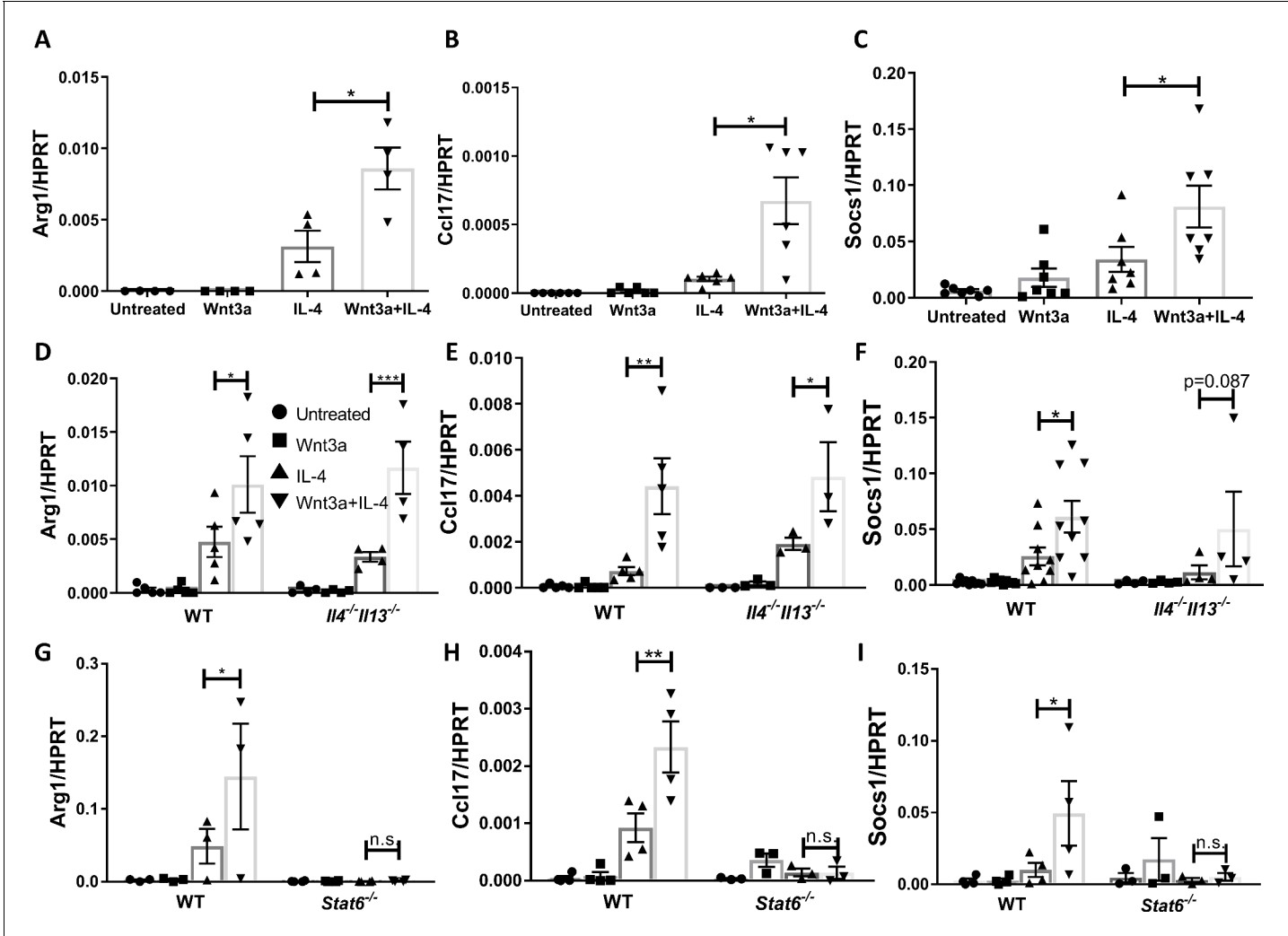

**Figure 5.** Wnt3a enhances macrophage IL-4-induced gene expression in vitro. Bone marrow-derived macrophages from WT (**A–C**), *Il4⁻/⁻Il13⁻/⁻* (**D–F**), or *Stat6⁻/⁻* (**G–I**) mice were treated with Wnt3a, IL-4, or their combination for 16 hr and expression of (**A/D/G**) *Arg1*, (**B/E/H**) *Ccl17*, and (**C/F/I**) *Socs1* was determined using qPCR. p-Values were determined via (**A–C**) one-way ANOVA or (**D–I**) two-way ANOVA compared with IL-4 treatment and Dunnett's multiple comparison test (with repeated measurements). *p<0.05; **p<0.01; ***p<0.001.

The online version of this article includes the following figure supplement(s) for figure 5:

**Figure supplement 1.** Macrophage responsiveness to Wnt stimulation in vitro.

that LGK974 treatment inhibited Wnt signaling in plaque macrophages, we excised them using laser capture microdissection (LCM) and examined the expression of Wnt-responsive genes. The data show decreased expression of *Lef1* and *Axin2* from LGK974 compared to vehicle-treated mice (***Figure 6—figure supplement 1H***).

Plaque area and macrophage content were significantly decreased in the control group compared to BL (***Figure 6C–F***). However, mice that received the Wnt inhibitor did not have a significant decrease in plaque or macrophage area (***Figure 6C–F***), despite equally low cholesterol levels (***Figure 6B***). Furthermore, we previously reported that resolving plaques are enriched with collagen, presumably because the content of matrix metalloprotease-producing macrophages decline (***Peled et al., 2017***). We again observed this increase in plaque collagen in the control resolution group, but it was not detected in the Wnt inhibitor group (***Figure 6F, G***). As noted above, atherosclerosis resolution is characterized by the change in the inflammatory state of plaque macrophages from pro-inflammatory to pro-resolving (***Rahman et al., 2017***; ***Ramsey et al., 2014***; ***Feig et al., 2012***). Consistent with this, plaque immunostaining showed that in control resolution there was

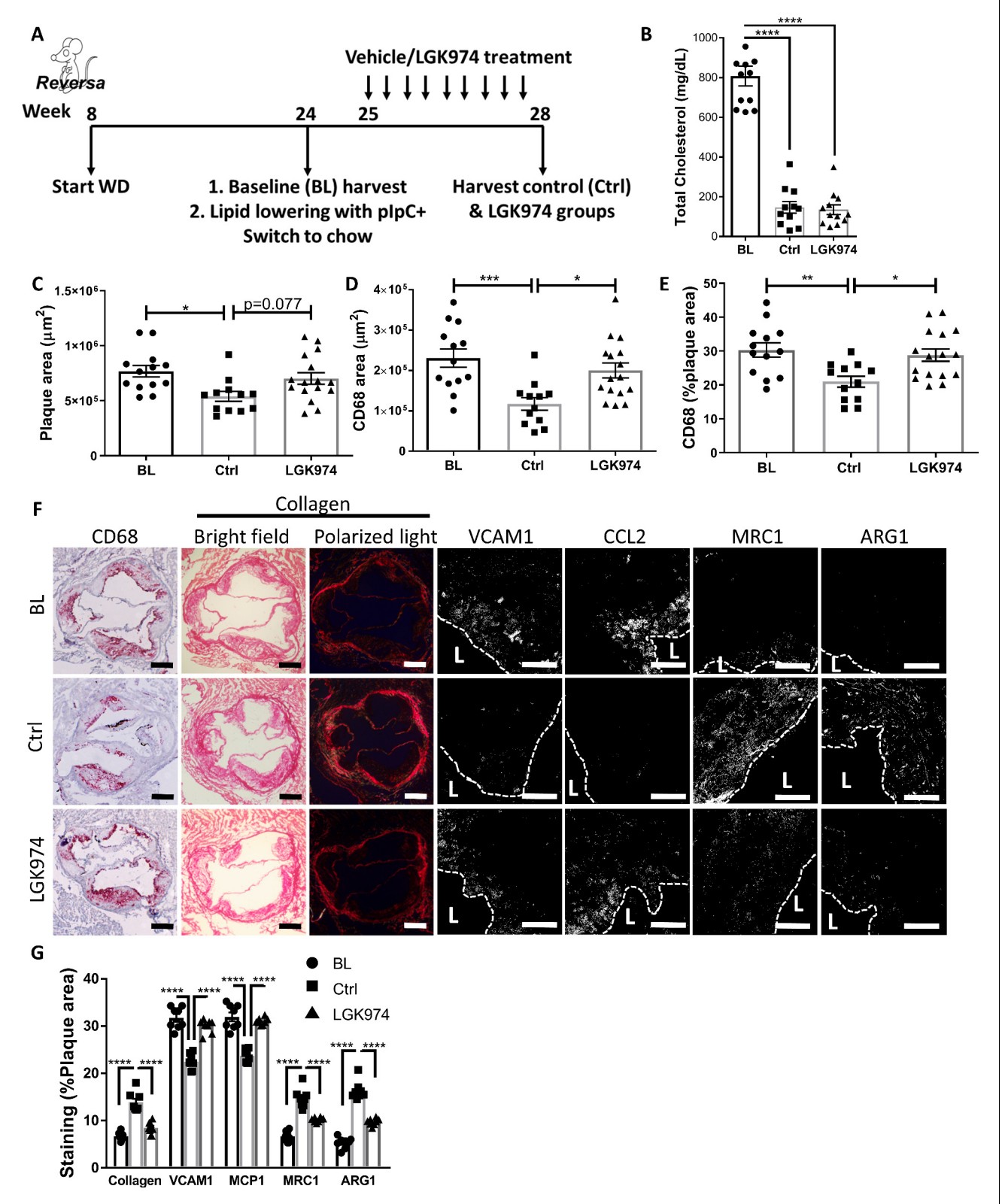

**Figure 6.** Wnt inhibition impairs atherosclerosis resolution and reparative macrophage polarization. (A) Experimental design. (B) Plasma cholesterol. Aortic root morphometric analysis for areas of (C) plaque, (D) CD68, and (E) %of plaque positive for CD68. (F) Representative images and (G) quantification of aortic roots stained for (from left to right) CD68, collagen (with picrosirius red, imaged in bright field and under polarized light), vascular cell adhesion molecule 1, C-C motif chemokine ligand 2, mannose receptor 1MRC1, and arginase 1. Scale bar, 200 μm for CD68 and collagen,
*Figure 6 continued on next page*

*Figure 6 continued*

50 µm for others. The dashed line demarcates plaque, and L stands for lumen. ****p<0.0001, ***p<0.001, **p<0.01, and *p<0.05 determined via one-way ANOVA comparing each group to the other groups, with Dunnett's multiple comparison test.

The online version of this article includes the following figure supplement(s) for figure 6:

**Figure supplement 1.** Effects of Wnt inhibition on general physiological parameters.

downregulation of the pro-inflammatory markers CCL2 and VCAM1, which were not attenuated in the LGK974 group. Additionally, compared to BL mice, in the control group there were increases in pro-resolving markers ARG1 and MRC1, which were not seen in the LGK974 group (*Figure 6F, G*). Overall, these data suggest that Wnt inhibition impairs atherosclerosis resolution and results in decreased accumulation of pro-resolving plaque macrophages.

## STAT3 is activated by Wnt3a and is required for the augmentation of IL-4

To investigate the molecular mechanisms underlying the enhanced response of BMDMs to Wnt3a and IL-4, we performed RNA-seq of cells treated for 16 hr with media (untreated) or with media containing Wnt3a, IL-4, or Wnt3a+IL-4. Principal component analysis (PCA) shows that the untreated BMDMs cluster with the IL-4-treated cells (*Figure 7A*), indicating a high similarity of transcriptional profiles (*Figure 7—figure supplement 1A*) before Wnt-mediated augmentation. Wnt3a-treated (−/+ IL-4) cells clustered together were distinct from the control and IL-4-treated cells (*Figure 7A*) and exhibited extensive transcriptional changes (*Figure 7—figure supplement 1A*). The data suggested a possible engagement of the JAK-STAT pathway in Wnt3a-treated cells as both Wnt3a and Wnt3a +IL-4-treated cells had the KEGG pathway term 'JAK-STAT signaling pathway' in the top seven most significantly enriched pathways (*Figure 7—figure supplement 1B*). Of the members of the JAK-STAT pathway, we focused on STAT3 because of its known anti-inflammatory functions (reviewed in *Hillmer et al., 2016*) and its being implicated in Wnt3a-induced enhancement of IL-4 responses of macrophages (*Feng et al., 2018*). Moreover, the STAT3 activating cytokines IL-6 and IL-10 have been reported to collaborate with IL-4 to enhance *Arg1* expression in macrophages (*Makita et al., 2015*; *Mauer et al., 2014*), as we also found (*Figure 7B, C*).

We next tested whether STAT3 is activated (as reflected by phosphorylation) by Wnt3a in BMDMs. The data show that STAT3 is not phosphorylated after a short exposure to Wnt3a (*Figure 7—figure supplement 1C*); however, after 16 hr of treatment, there was substantial STAT3 phosphorylation (*Figure 7D*). Furthermore, the STAT3 target gene *Socs3* was upregulated following Wnt3a treatment (*Figure 7E*). To directly examine the involvement of STAT3 in the enhanced expression of pro-resolving genes following Wnt3a treatment, we knocked-down (KD) *Stat3* mRNA in BMDMs using siRNA and treated these cells with Wnt3a, IL-4, or Wnt3a+IL-4. The siRNA decreased *Stat3* transcript levels by ~80% (*Figure 7—figure supplement 1D*). Results show that *Stat3* KD attenuated *Arg1* expression in response to Wnt3a+IL-4 (*Figure 7F*), suggesting that STAT3 is required for the enhancement of IL-4 responses by Wnt3a.

To explore whether Wnt inhibition affects STAT3 activity in vivo, aortic root sections from control and LGK974-treated mice (*Figure 6A*) were stained for STAT3, and its nuclear localization was determined. Images and their analyses show less nuclear STAT3 localization in plaques of mice treated with LGK974 compared to the control group (*Figure 7G*). Collectively, these data suggest that Wnt enhances the response to IL-4 via STAT3 activation.

We next aimed at understanding how Wnt signaling promotes STAT3 activation. Since STAT3 is most commonly activated downstream of cytokine receptors, together with our observation that Wnt3a promotes STAT3 phosphorylation 16 hr post treatment, but not in early time points, we postulated that Wnt3a induces the production and secretion of a factor or factors that accumulate in the conditioned media (CM) and activate STAT3 in an autocrine/paracrine manner. To test this hypothesis, CM were generated from BMDMs treated with Wnt3a for 16 hr. Naive BMDMs were then treated with the CM or Wnt3a itself for 15, 30 and 60 min, and STAT3 phosphorylation was examined. The data show that the CM caused STAT3 phosphorylation after 15 min, while Wnt3a itself failed to do so (*Figure 7—figure supplement 1E*). Furthermore, to examine whether Wnt CM augments pro-resolving gene expression, BMDMs were treated with IL-4 for 2.5 hr either after 16 hr of

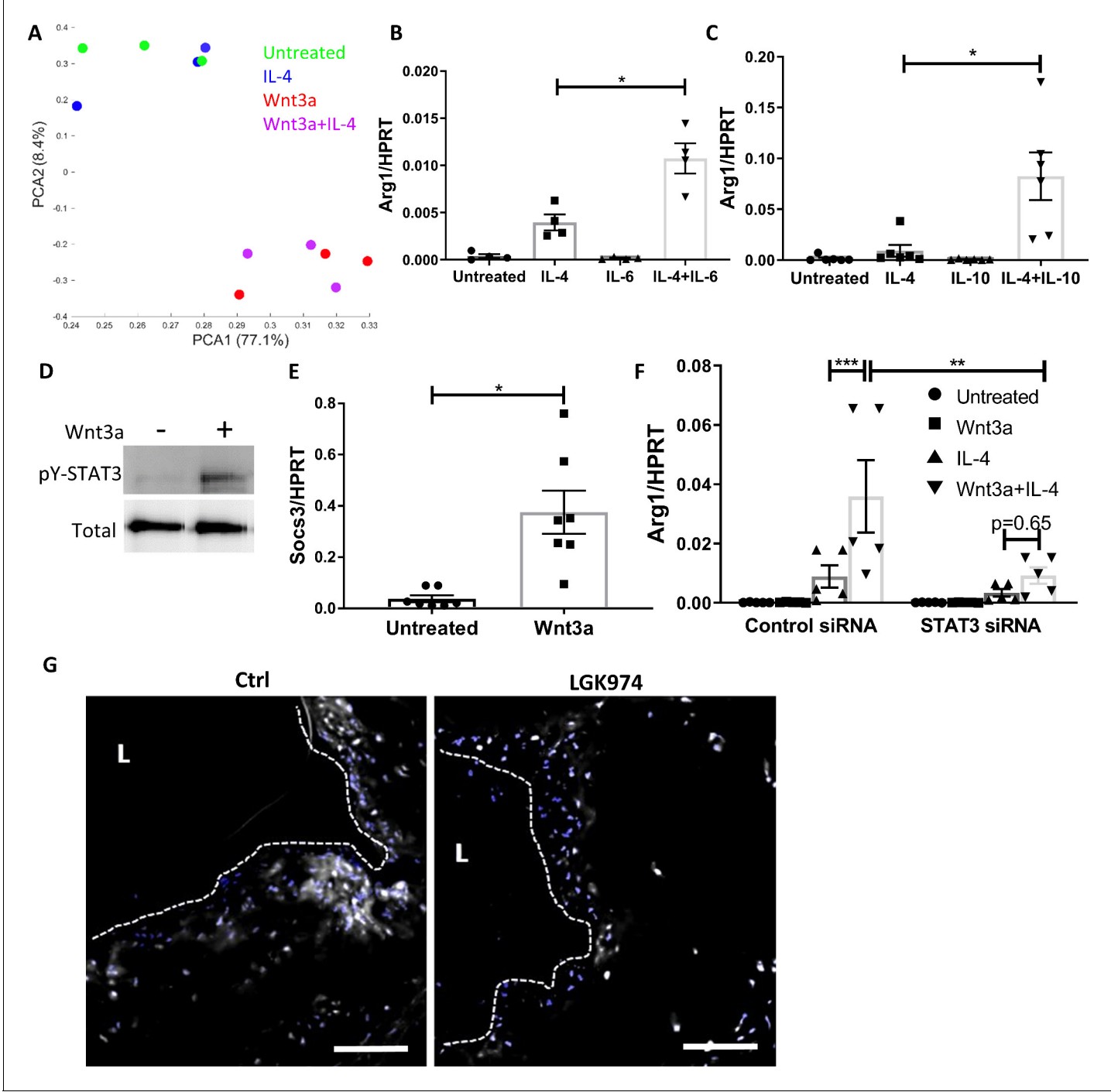

**Figure 7.** Wnt3a promotes STAT3 activation, which is required for the enhanced pro-resolving phenotype. (**A**) Principal component analysis of RNA-seq data acquired from bone marrow-derived macrophages (BMDMs) treated with Wnt3a, IL-4, or their combination. (**B, C**) *Arg1* expression of BMDMs treated with IL-4 and (**B**) IL-6 or (**C**) IL-10. (**D**) Representative western blot image for phosphorylates (pY) and total STAT3 16 hr following treatment of BMDMs with Wnt3a (one out of three independent experiments). (**E**) BMDMs were treated with Wnt3a for 16 hr, and *Socs3* expression was determined using qPCR. p-Value was determined by a paired *t*-test (two-tailed). (**F**) BMDMs were treated with control or STAT3 siRNA, followed by treatment with Wnt3a, IL-4, or their combination and *Arg1* expression determined using qPCR. p-Values were determined via (**B, C**) one-way or (**F**) two-way ANOVA compared with IL-4 treatment. Between siRNA groups, p-values were determined via two-way ANOVA comparing same Wnt3a/IL-4 treatments and Dunnett's multiple comparison test (with repeated measurements). ***p<0.001, **p<0.01, and *p<0.05. (**G**). Representative images of aortic roots from control (Ctrl) or LGK974-treated mice (as in *Figure 6A*) stained for nuclei (blue) and STAT3 (gray). Scale bar, 100 μm.

The online version of this article includes the following figure supplement(s) for figure 7:

*Figure 7 continued on next page*

*Figure 7 continued*

**Figure supplement 1.** STAT3 involvement in the Wnt3a-mediated enhancement of IL-4 response.

Wnt3a treatment or in combination with Wnt CM. Similar to our previous results, co-treatment of BMDMs with Wnt3a, as well as Wnt CM, in combination with IL-4 promoted enhanced *Arg1* expression compared with IL-4-only treatment (*Figure 7—figure supplement 1F*). To confirm that the augmentation in *Arg1* expression is not due to residual Wnt3a, CM was treated with a Wnt3a-blocking antibody prior to its exposure to naïve BMDMs. The data show that the CM retained its IL-4 augmentation ability (*Figure 7—figure supplement 1F*), suggesting that Wnt3a stimulated the production and/or secretion of a factor/s other than itself that, in turn, activated STAT3.

As a first step in the investigation of the Wnt3a-stimulated secreted factor that activates STAT3, we mined our RNA-seq dataset (*Figure 7A*) for soluble proteins with known links to STAT3 activation that showed differential expression following Wnt3a treatment. This showed differential expression of transforming growth factor β (TGFβ) and C-X-C motif chemokine receptor 3 (CXCR3) in Wnt3a-treated cells. Next, TGFβ and CXCR3, as well as the main receptors upstream of STAT3 activation (glycoprotein 130 [gp130], IL-10 receptor [IL10R], and hepatocyte growth factor receptor [HGFR]), were inhibited using specific blocking antibodies, concomitant with treatment of the BMDMs with IL-4 alone or a combination of IL-4 and Wnt3a. *Arg1* expression was not attenuated by blocking any of the aforementioned proteins in Wnt3a+IL-4-treated cells (*Figure 7—figure supplement 1G*), indicating that these factors do not play a role in the enhanced pro-resolving gene expression following Wnt3a treatment.

## Wnt3a induces the production of PGE$_2$, which causes the enhanced response to IL-4

Because we could not find a potential secreted protein to activate STAT3 following Wnt3a treatment, together with experiments showing that boiled Wnt CM retained its ability to augment *Arg1* expression (data not shown), we postulated that the Wnt3a-stimulated secreted factor is a lipid. Thus, targeted liquid chromatography-tandem mass spectrometry (LC-MS)/MS metabololipidomics analysis of Wnt3a CM was performed. We were particularly interested in inflammation-resolving lipid mediators (representative MS/MS fragmentation spectra are shown in *Figure 8—figure supplement 1A, B*). We found that several lipid mediators were upregulated in CM of Wnt3a-treated BMDMs (*Figure 8A*), notably at the time when CM enhances the response to IL-4. Specifically, there was a global increase of arachidonic acid metabolites, such as TXB$_2$, PGE$_2$, PGD$_2$, PGF$_{2a}$, peaking 16 hr post-Wnt3a treatment (respective fold changes of 18, 58, 101, and 5, *Figure 8A*).

Of the upregulated metabolites, we focused on PGE$_2$ because of its function as an amplifier of IL-4 responses (*Sanin et al., 2018*; *Gao et al., 2016*; *Luan et al., 2015*; *Heffron et al., 2021*; *Figure 8—figure supplement 1C*) and its role as a STAT3 activator in various cell types (*Jw et al., 2019*; *Chun et al., 2010*; *Wang et al., 2015*; *Halpern et al., 2019*). For example, Sanin et al. recently examined the relationship between PGE$_2$ and IL-4 to regulate pro-resolving gene expression and have performed RNA-seq of BMDMs co-treated with IL-4 and PGE$_2$ (*Sanin et al., 2018*). Thus, we compared those data to our RNA-seq data of BMDMs co-treated with Wnt3a and IL-4. The data show a high degree of correlation in gene expression between BMDMs treated with Wnt3a+IL-4 and PGE$_2$+IL-4, with $r^2 = 0.481$ and p-value<$2.34^{-75}$ (*Figure 8B*), despite marked differences in the IL-4 concentrations used in these experiments (0.1 and 20 ng/mL, *Sanin et al., 2018*, respectively).

Because the canonical Wnt pathway was previously shown (e.g., *Araki et al., 2003*; *Nunez et al., 2011*) to regulate *Ptgs2* expression (the enzyme that metabolizes arachidonic acid to PGH$_2$, the PGE$_2$ precursor), we examined its levels following Wnt3a stimulation in BMDMs. The results show increased *Ptgs2* expression upon Wnt3a treatment, as well as upregulation of the next enzyme in the PGE$_2$ production pathway, *Ptges* (*Figure 8C*). In addition to the increase in these enzymes that produce PGE$_2$, Wnt3a also decreased the expression of *Hpgd*, which inactivates PGE$_2$ and serves as a negative regulator of its levels (*Figure 8C*).

To directly assess whether PGE$_2$ is the secreted factor in Wnt3a CM that cooperates with IL-4, we employed two strategies. First, Wnt CM was treated with a PGE$_2$-blocking antibody and then added to naive BMDMs together with IL-4 for 2.5 hr. While blocking of Wnt3a itself in Wnt CM did not alter

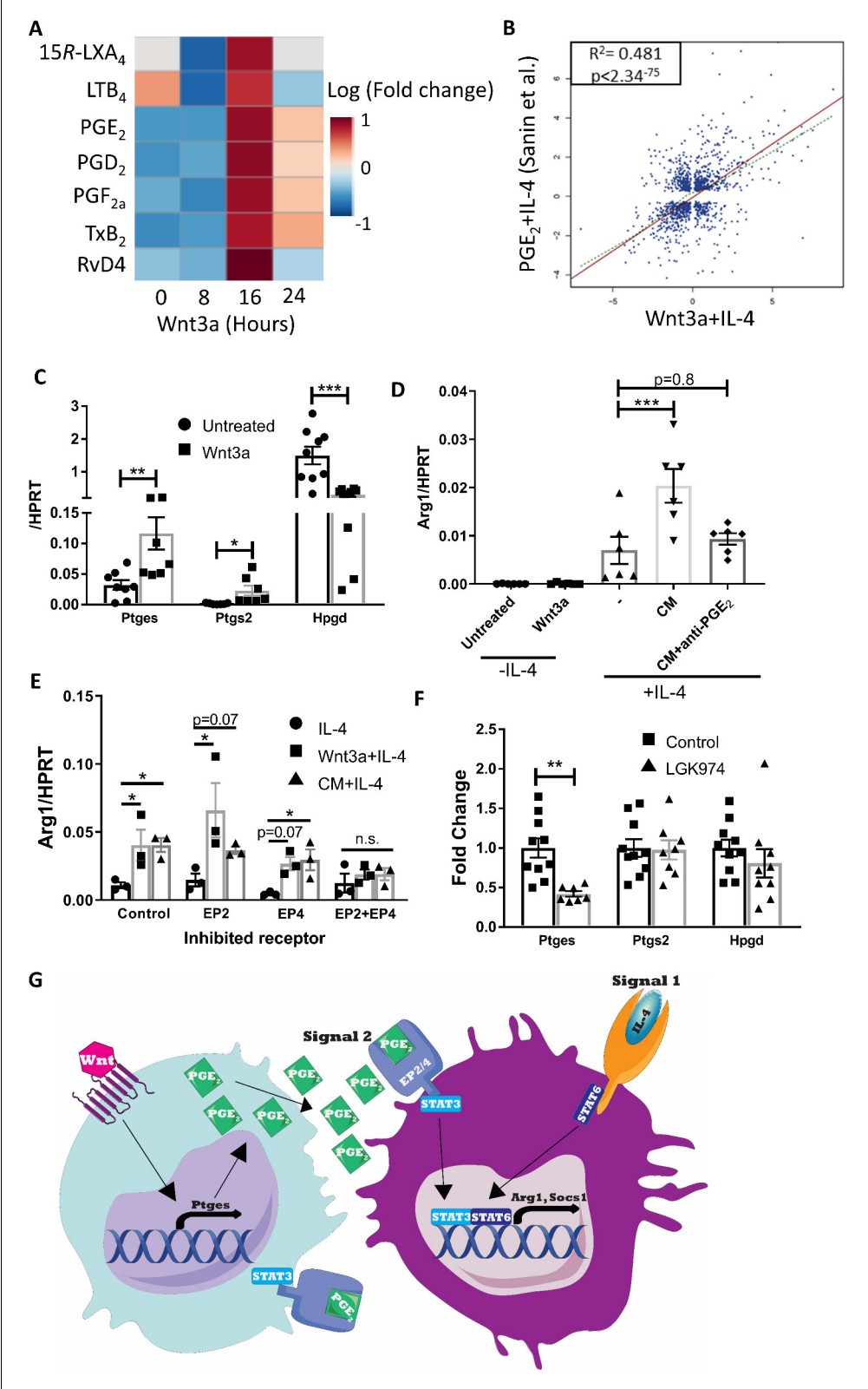

**Figure 8.** Wnt3a promotes PGE$_2$ production that enhances the response to IL-4. (**A**) Bone marrow-derived macrophages (BMDMs) were treated with Wnt3a for the indicated times and conditioned media (CM) analyzed using LC-MS/MS. Relative abundance to untreated cells shown. (**B**) Correlation between genes differentially expressed (adjusted p-value<0.05) in Wnt3a+IL-4 (from Weinstock et al.) or PGE$_2$+IL-4 (from Sanin et al.). (**C**) BMDMs were treated with Wnt3a for 16 hr and expression of *Ptges*, *Ptgs2*, and *Hpgd* determined with qPCR. (**D**) Wnt CM was obtained (same experiment as

*Figure 8 continued on next page*

Figure 8 continued

*Figure 7—figure supplement 1F*). Naïve BMDMs were treated with IL-4 or CM+IL-4 for 2.5 hr. CM was treated with a PGE$_2$ blocking antibody prior to their exposure (with IL-4) to naïve BMDMs. *Arg1* expression was determined. (E) BMDMs were treated with Wnt3a for 16 hr and an additional 2.5 hr with IL-4 or with CM+IL-4 for 2.5 hr. Cells were concomitantly treated with either DMSO (Dimethyl sulfoxide; control), EP2 inhibitor, EP4 inhibitor, or their combination for the duration of treatment with Wnt3a or CM. *Arg1* expression was evaluated. p-Values were determined via (D) one-way and (E) two-way ANOVA, compared with IL-4 treatment, and Dunnett's multiple comparison test (with repeated measurements). (F) Expression of *Ptges*, *Ptgs2,* and *Hpgd* in aortic root plaque macrophages of control (Ctrl) and LGK974 groups (as in *Figure 6A*). (C, F) p-Values were determined by an unpaired *t*-test (two-tailed). \*\*\*p<0.001, \*\*\*p<0.01, and \*p<0.05. (G) Schematic model of findings.

The online version of this article includes the following figure supplement(s) for figure 8:

**Figure supplement 1.** Prostanoids measurements and PGE$_2$ cooperation with IL-4.

the augmented *Arg1* expression (*Figure 7—figure supplement 1F*), PGE$_2$ blockade completely abrogated the enhancement in gene expression (*Figure 8D*). In a complementary experiment, we inhibited the PGE$_2$ receptors EP2 and EP4 prior to exposure of BMDMs to Wnt CM and IL-4. The results showed that control BMDMs had increased *Arg1* expression upon treatment with Wnt3a+IL-4 or CM+IL-4 (compared with IL-4 alone), with similar responses in BMDMs pre-treated with either EP2 or EP4 inhibitors (*Figure 8E*). However, BMDMs in which both receptors were inhibited simultaneously did not show enhanced *Arg1* expression upon Wnt3a+IL-4 or CM+IL-4 treatment, but rather responded similarly to IL-4-only-treated cells (*Figure 8E*).

Finally, to determine whether Wnt inhibition in vivo may affect PGE$_2$ production, we examined the expression of *Ptgs2*, *Ptges,* and *Hpgd* in macrophages LCM-excised from plaques of control or LGK974-treated mice (as in *Figure 6—figure supplement 1I*). Although expression of *Ptgs2* and *Hpgd* did not differ between the two groups, the expression of *Ptges* was reduced by 58% in the LGK974 group compared to controls (*Figure 8F*). Taken together, our data suggest that macrophages activated by the Wnt pathway produce and secrete PGE$_2$, activating STAT3 in an autocrine/paracrine fashion, which subsequently cooperates with IL-4/STAT6 (*Figure 8G*). In atherosclerosis, this mechanism primes the response of plaque macrophages to IL-4 that accumulates in plaques during disease progression and together induce resolution of atherosclerosis.

## Discussion

We have previously shown that aggressive lowering of ApoB-containing lipoproteins leads to resolution of atherosclerosis in mice, a process characterized by decreases in plaque macrophages and their expression of inflammatory genes, and by increased expression of pro-resolving genes. Among these is *Arg1*, which, besides depriving substrate to damaging iNOS (*Munder, 2009*), has been recently shown to be essential for the clearance of multiple apoptotic cells (*Yurdagul et al., 2020*), an important process in inflammation resolution in plaques and in general (*Libby et al., 2014*). The expression of *Arg1* and other pro-resolving genes (i.e., *Ccl17* and *Socs1*) is regulated by STAT6 (*Van Dyken and Locksley, 2013*; *Rutschman et al., 2001*), which is concordant with our recent demonstration that following lipid lowering monocytes recruited to plaques in a STAT6-dependent manner mediate resolution (*Rahman et al., 2017*).

We now extend these findings to show the following: (1) IL-4 promotes the resolution of atherosclerosis, though its levels in progressing or resolving plaques are low and stable; (2) the source of IL-4 required for disease resolution is from cells that accumulate in plaques during disease progression and not cells recruited following lipid lowering; (3) although the IL-4/STAT6 pathway is needed for disease resolution, it requires Wnt signaling to potentiate full IL-4 responsiveness; (4) loss of Wnt signaling in plaque macrophages is associated with impaired atherosclerosis resolution in mice, consistent with premature coronary disease in people with a loss-of-function mutation in the Wnt co-receptor *Lrp6* (*Mani et al., 2007*), and with the lower level of gene expression of Wnt targets in symptomatic versus asymptomatic human plaques; and (5) Wnt signaling induces the production and secretion of PGE$_2$ and activation of STAT3, which are required for the enhancement of IL-4-induced effects on expression of genes associated with inflammation resolution. These findings come from integrating results from multiple mouse atherosclerosis studies, with biochemical and bioinformatic data from in vivo and in vitro systems, including from human plaques.

These results suggest that IL-4 alone is not sufficient for atherosclerosis resolution and that Wnt signaling cooperates with IL-4 to enhance its induction of the expression of genes associated with inflammation resolution. Of note, the inability of IL-4/13 to independently mount a sufficient response in macrophages to promote tissue repair, which atherosclerosis is an example of, has been recently shown in other contexts, including in the lungs after helminth infection or in the gut after induction of colitis (*Bosurgi et al., 2017*). Rather, recognition of apoptotic cells through the receptors MER proto-oncogene, tyrosine kinase (MerTK) and Axl receptor tyrosine kinase together with IL-4 was needed for this induction in these two models (*Bosurgi et al., 2017*). Another relevant study showed that IL-4/13 collaborate with local tissue enhancers that amplify their response upon helminth infection (*Minutti et al., 2017*). Deletion of these local enhancers caused a drastic impairment in the inflammatory response against parasites, although IL-4/13 levels were unaffected.

It is plausible that the need for tissue enhancers arises from relatively low tissue IL-4 concentrations, which is supported by our finding that plaques contain ~100 pg/mL IL-4 (*Figure 3A*), a concentration that is 100–200-fold lower than what is used in typical in vitro studies (e.g., *Minutti et al., 2017*; *Bosurgi et al., 2017*; *Sanin et al., 2018*). Overall, the results from the other systems bear a striking resemblance to the scheme in *Figure 8G*, in which we posited that for full IL-4/STAT6 responsiveness plaque macrophages require a second signal, which, in the case of atherosclerosis, is engagement of the Wnt pathway that promotes the production of $PGE_2$ and STAT3 activation. Further support for a 'two factor' model of IL-4 responsiveness, in general, and Wnt being one such factor, in particular, also comes from a model of kidney injury and fibrosis, in which Wnt3a caused the activation of STAT3 and enhanced IL-4 responses (*Feng et al., 2018*).

One caveat of this study is that it is possible that plaque progression is attenuated by the absence of IL-4 and IL-13, thereby making resolution harder to observe. Indeed, some previous studies reported less plaque progression in $Apoe^{-/-}Il4^{-/-}$ (*Davenport and Tipping, 2003*) and $Ldlr^{-/-}Il4^{-/-}$ (*King et al., 2002*) compared to IL-4-sufficient mice. In contrast, an elegant study from Binder and colleagues *Cardilo-Reis et al., 2012* demonstrated that *Il13*-deficiency *enhanced* atherosclerosis progression and that pharmacological treatment with IL-13 inhibited macrophage accumulation and promoted pro-resolving macrophage polarization in plaques. Notably, in the present study, substantial plaques are formed in $Il4^{-/-}Il13^{-/-}$ mice (average plaque size of 319,293 $\mu m^2$; *Figure 1C*), and our previously published data have shown atherosclerosis regression in significantly smaller plaques (<200,000 μm, *Basu et al., 2018*; *Sharma et al., 2020*). Taken together, these findings support our conclusion that IL-4 is required for resolution of atherosclerosis of plaques of the type we studied.

Wnt proteins are evolutionary ancient and are highly abundant and diverse. Despite many Wnt proteins, there are common transcriptional targets, allowing the assessment of the pathway activity with transcriptomic data. As noted earlier, we reported the upregulation of Wnt-responsive genes in resolving plaque macrophages in two independent mouse models of atherosclerosis resolution (*Ramsey et al., 2014*). In the present study, we show in a third model, but now on a single-cell level, that resolving plaque macrophages have higher expressions of Wnt-regulated genes (*Figure 4A–C*). That this may be relevant to human atherosclerosis is supported by the data in *Figure 4D–F*, which indicate lower activity of the Wnt pathway in macrophages from symptomatic human plaques, compared with asymptomatic ones, and increased Wnt-responsive gene expression in PBMCs from humans placed on LDL cholesterol-lowering therapy after an acute coronary syndrome. In particular, after 3 months of treatment, there was a significant inverse correlation between the expression of the canonical Wnt-target gene, *Axin2*, and plaque size. The studies in mice (*Figure 6*) and humans (*Figure 4D–F* and the *Lrp6* mutation (*Mani et al., 2007*) in the aggregate strongly support an inverse relationship between Wnt activity and atherosclerosis severity.

Turning to STAT3, our results demonstrate that it is downstream of Wnt and collaborates with IL-4/STAT6 to enhance IL-4-induced gene expression (*Figure 7F*). This is in accordance with the aforementioned report showing enhanced response of BMDMs to IL-4 following Wnt3a treatment via activation of STAT3 in a model of kidney injury (*Feng et al., 2018*), as well as with the suggestion that STAT3 and STAT6 cooperate to increase *Arg1* gene expression (reviewed in *Pourcet and Pineda-Torra, 2013*). We extended the findings in vitro to show in vivo that STAT3 activation appears to be deficient in plaques upon Wnt inhibition (*Figure 7G*). STAT3 is a particularly interesting STAT family member due to its important contributions to both inflammatory and anti-inflammatory responses. The mechanisms by which STAT3 induces such diverse responses are still unclear, but will be crucial to understand in order to target the intended function. One possibility is that STAT3 works with

other transcription factors that are engaged downstream of various receptors. For instance, it had been previously shown that STAT3 (activated by IL-6) and STAT6 (activated by IL-4) cooperate to enhance expression and secretion of cathepsins, to promote polarization of tumor-associated macrophages (*Yan et al., 2016*) and enhance IL-4-induced gene expression (*Makita et al., 2015*; *Fernando et al., 2014*; *Figure 7B, C*). Our findings appear to be in a similar vein, in that Wnt3a-induced STAT3 activation collaborated with IL-4-induced STAT6 activation to enhance responsiveness to IL-4.

Our data further suggest that STAT3 is activated following Wnt stimulation via $PGE_2$ since the latter is produced by Wnt3a-treated BMDMs (*Figure 8A*) and has been shown to promote the phosphorylation of STAT3 in various cell types (*Jw et al., 2019*; *Chun et al., 2010*; *Wang et al., 2015*; *Halpern et al., 2019*). Wnt signaling promotes the production of $PGE_2$ through transcriptional regulation of *Ptgs2* (*Araki et al., 2003*; *Nunez et al., 2011*), with similar results shown in *Figure 8C*. However, Wnt transcriptional control of other enzymes that regulate $PGE_2$ levels is less known. Our data show that in vitro, Wnt3a promoted the expression not only of *Ptgs2*, but also *Ptges*, and decreased the expression of *Hpgd* (*Figure 8C*). Since PTGS2 and PTGES are needed for $PGE_2$ synthesis and HPGD degrades $PGE_2$, the net effect of Wnt-induced gene expression would be expected to increase $PGE_2$ concentrations, which we observed (*Figure 8A*). Interestingly, in plaque macrophages, of the $PGE_2$-regulating enzymes, only the expression of *Ptges* was decreased upon Wnt inhibition, indicating a potentially relatively greater contribution of this enzyme in vivo in Wnt-mediated atherosclerosis resolution. Further supporting that $PGE_2$ is an important intermediary between Wnt signaling and the amplification of IL-4 responses were the experiments in which $PGE_2$ blockade or inhibition of its receptors abrogated this amplification (*Figure 8D, E*). $PGE_2$ itself was previously shown to amplify IL-4-induced gene expression in macrophages (*Sanin et al., 2018*; *Gao et al., 2016*; *Luan et al., 2015*).

To place the present results in a wider context, the involvement of $PGE_2$ receptors in the development of atherosclerosis was also studied previously. Initially, it was reported that inhibition of $PGE_2$ responses via deletion of its receptor EP4 (but not EP2) decreased early atherosclerosis burden (*Babaev et al., 2008*). However, this study did not report a significant change in plaque macrophage content in EP2 or EP4-deficient conditions. Later reports showed that EP4 deficiency in bone marrow (*Tang et al., 2011*) or myeloid cells (*Vallerie et al., 2016*) caused an increase in plaque inflammation and macrophage content, without affecting diabetic atherosclerosis. These data, together with results presented here, indicate that engagement of the $PGE_2$ receptor EP4 in macrophages may be beneficial not only for dampening atherosclerosis progression, but also enhancing disease resolution. However, whether direct activation of the receptors for $PGE_2$ will enhance atherosclerosis resolution merits additional investigation.

Also deserving of further investigation is the source of Wnt. Moreover, though from our data in vitro, it is reasonable to propose that $PGE_2$ production following Wnt stimulation acts in an autocrine or paracrine manner. Extrapolation to the situation in vivo has not been established, but the human data presented in *Figure 4E, F* suggest a systemic increase in Wnt responsiveness since circulating immune cells showed higher expression of Wnt-responsive genes in patients whose plaques were regressing as assessed by imaging. Furthermore, based on our mouse results, it is plausible that Wnt production or its secretion had increased in these patients upon lipid lowering.

In conclusion, the present study provides new insights into the molecular pathways involved in resolution of atherosclerosis, which remains an important clinical goal. While our previous studies pointed to IL-4 as a polarizer of plaque macrophages to drive them towards inflammation resolution, we were surprised to discover that this cytokine is required, but not sufficient, and that Wnt signaling is needed as well. Given the human data cited above, the clinical relevance of our findings is also plausible. Thus, the demonstrated molecular mechanism suggests a potential therapeutic strategy to improve the resolution of atherosclerosis after lipid lowering, namely, by co-activation of the IL-4 and $PGE_2$ pathways in plaque macrophages.

## Materials and methods

### Study design

This study aimed at determining molecular mechanisms upstream of STAT6 that promote atherosclerosis resolution. Mouse studies of atherosclerosis resolution were performed on the basis of LDLr deficiency (*Ldlr*[-/-], *Pcsk9*-AAV (*Peled et al., 2017*; *Bjørklund et al., 2014*; *Roche-Molina et al., 2015*) and Reversa *Feig et al., 2011a*; *Lieu et al., 2003*), with WD (100244; Dyets Inc) feeding for 16–20 weeks, followed by aggressive cholesterol lowering (using ApoB-ASO for *Ldlr*[-/-] and *Pcsk9*-AAV or poly I:C injections for Reversa; *Feig et al., 2011a*; *Lieu et al., 2003*) with chow diet feeding, according to the protocol (number IA16-00494) approved by the Institutional Animal Care and Use Committee of the New York University School of Medicine. In the aortic transplant model (*Rahman et al., 2017*; *Chereshnev et al., 2003*), aortic arches from atherosclerotic (*Ldlr*[-/-]/*Apoe*[-/-]) mice were intrapositioned with the abdominal aorta in recipient mice (*Il4*[-/-]*Il3*[-/-]/*Il4*[-/-], maintained on a chow diet), and blood flow was directed through the graft. For in vivo Wnt inhibition studies, mice were administered 3 mg/kg LGK974 (*Liu et al., 2013*) (S7143; Celleckchem) three times/week by oral gavage. For all studies, mice were randomized into the groups at the BL time point. Researchers were not blinded to the identity of the study groups, with blinding prior to data analysis. Based on a power analysis (performed as described on https://stats.idre.ucla.edu/other/mult-pkg/seminars/intro-power/), we typically enroll 15 mice to have at minimum 10 mice/group; this is estimated to give 90% power to detect differences greater than 20% in atherosclerotic plaque parameters with ≈15% standard deviation between wild type and test mice based on two-sided two-sample *t*-test with $\alpha = 0.05$. In some cases, fewer mice were used in order to have enough tissues for all of the sample types needed.

### Animals

C57BL/6, *Il4*[-/-], *Stat6*[-/-], *Apoe*[-/-], *Ldlr*[-/-], and Reversa mice were purchased from The Jackson Laboratory. *Il4*[-/-]*Il13*[-/-]mice (C57BL6/J background) were provided by Dr. Thomas Wynn (then at the National Institutes of Health). IL-4/eGFP mice were provided by Dr. Richard Locksley (University of California San Francisco). Male and female mice at ages 6–10 weeks were used.

### Pcsk9 AAV model

This new atherosclerosis resolution model was recently described (*Peled et al., 2017*; *Bjørklund et al., 2014*; *Roche-Molina et al., 2015*; *Sharma et al., 2020*). Briefly, mice were injected intraperitoneally with an AAV that expresses a hyperactive form of *Pcsk9* (AAV.8TBGmPCSK9D377Y, Penn Vector Core), under the regulation of a liver-specific promoter, at $1 \times 10^{12}$ viral particles/mouse. After 16–20 weeks of WD feeding, lipid lowering was achieved by discontinuing the WD feeding and bi-weekly i.p. injections of 50 mg/kg ASO to ApoB, kindly provided by Ionis Pharmaceuticals. Mice were sacrificed after 3 weeks of ApoB ASO treatment and their tissues collected.

### Human specimens

For the data displayed in *Figure 4D*, they were based on the initial analysis of human atherosclerosis samples as previously reported (*Fernandez et al., 2019*). Subjects gave informed written consent, following the approval by the Institutional Review Board of the Icahn School of Medicine at Mount Sinai (IRB 11-01427). For the data displayed in *Figure 4E, F*, the examination of atherosclerosis resolution features in humans was described previously (*Alkhalil et al., 2018*). Ethical approval was obtained from National Research Ethics Services (NRES) and Oxford R and D committee prior to commencement of the study. All patients provided written informed consent prior to participation in this study, which included the collection of peripheral blood cells and their analysis.

### Plaque morphometrics and immunohistochemistry

Hearts were removed after perfusion at physiological pressure of saline containing 10% sucrose, embedded in OCT, and frozen. Hearts were sectioned through the aortic root (6 µm) and stained for CD68 (MCA1957; AbD Serotec) to detect macrophages, active-β-catenin (8814; Cell Signaling Technology), STAT3 (8768; Cell Signaling Technology), MRC1 (MCA2235; AbD Serotec), ARG1 (sc-20150; Santa Cruz), VCAM1 (ab134047; Abcam), and CCL2 (505908; BioLegend). Picrosirius red was

used for collagen staining, and imaging done using polarizing light microscopy. Image analysis was performed with ImagePro Plus 7.0 software (Media Cybernetics) and ImageJ (National Institute of Health *Schneider et al., 2012*).

## Flow cytometry

Aortic arches were digested as previously described (*Tang et al., 2015*), after their dissection following perfusion with PBS. Single-cell suspensions were added a live/dead cell staining (L23105; Invitrogen) and PE-Cy7 anti-CD45 antibody (103132; Biolegend) and analyzed using the LSRII cytometer (BD Biosciences).

## Plaque cytokine measurement

Plaques from *Ldlr*^-/- mice fed WD for 20 weeks and treated or not with ApoB ASO for 3 weeks were carefully scraped out from the aortic wall, placed in a tube containing 100 µL PBS and protease/phosphatase inhibitor (A32961, Thermo Fisher Scientific) and weighed. Plaques were manually homogenized until no pieces were apparent (~2 min). Concentrations of CCL2, IL-4, and IL-13 in suspensions were measured using a custom mouse LEGENDplex assay (Biolegend).

## Bone marrow-derived macrophages

Bone marrow was harvested from the femur and tibia of 6–20-week-old mice and treated with red-blood cell lysis buffer (R7757; Sigma-Aldrich) for 5 min. After washing, cells were seeded in DMEM media supplemented with 10% FBS, 1% penicillin/streptomycin, and 10 ng/mL macrophage colony-stimulating factor (m-CSF) (315-02; PeproTech). Cells were supplemented with media every other day and used for experiments between day 5 and 10 post harvest. Cells were typically treated with 100 ng/mL Wnt3a (1324-WN-010/CF; R&D Systems), 100 pg/mL IL-4, 5 ng/mL IL-6, and 2.5 ng/mL IL-10 (214-14, 216-16, and 210-10, respectively; Peprotech) for 16 hr, unless indicated otherwise. Experiments with Wnt CM or 100 pg/mL PGE$_2$ (14010; Cyaman Chemical) include treatment with IL-4 for 2.5 hr. For experiments with siRNA, BMDMs were transfected overnight with 25 nM scrambled (control) or STAT3 siRNA (L-040794-01; Dharmacon) using Lipofectamine RNAiMax Transfection reagent (13778075; Thermo Fisher Scientific), and started on Wnt3a and IL-4 treatments 48 hr after transfection. Blocking antibodies and protein inhibitors were used in the following concentrations: 0.5 µg/mL anti-TGFβ (521707; Biolegend), 0.1 µg/mL anti-gp130 from (MAB4682; R&D Systems), 1 µg/mL anti-HGFR (AF527; R&D Systems), 10 µg/mL anti-IL10R (112707; Biolegend), 0.1 µM AMG487 CXCR3 inhibitor (4487; Tocris Bioscience), 1 µg/mL anti-Wnt3a (MAB9025; R&D Systems), 100 nM anti-PGE$_2$ (10009814; Cayman Chemical), 30 µM AH6809 EP2 inhibitor (14050; Cayman Chemical), and 30 µM GW627368X EP4 inhibitor (10009162; Cayman Chemical).

## Laser capture microdissection

Under strict RNAse-free conditions, aortic root sections were stained with hematoxylin-eosin and captured. RNA was then isolated using the PicoPure Kit (KIT0204; Thermo Fisher Scientific), and quality and quantity were determined using an Agilent 2100 Bioanalyzer (5067-1513; Agilent Technologies). RNA was converted to cDNA and amplified using the WT-Ovation Pico RNA Amplification Kit (3302-12; NuGEN) (*Ramsey et al., 2014*; *Feig et al., 2012*; *Trogan et al., 2002*).

## Real-time qPCR

Total RNA was isolated using TRIzol reagent (15596-018; Life Technologies) and Direct-zol RNA Mini-iPrep columns (R2052; Zymo Research) and quantified using Nanodrop 2000 (Nanodrop Products). Quantitative real-time PCR was performed using FAST SYBR Green Master Mix (4385612; Applied Biosystems) on the ABI PRISM 7300 sequence detection system (Applied Biosystems). The mRNA levels were normalized to *Gapdh* or *Hprt* as reference genes. Primer sequences are available in *Supplementary file 1*.

## Human PBMC analysis of Wnt-responsive genes

RNA was isolated from PBMCs collected from patients in the study in reference (*Alkhalil et al., 2018*), and the expression of Wnt target genes (*Axin2, Ddx3x, Wls, Senp2, Ctnnb1, Csnk1a1,* and

*Amfr*) was measured by RT-PCR, with 18S RNA as a reference. Imaging of carotid atherosclerotic plaques was performed as described in *Alkhalil et al., 2018*.

## RNAseq analysis

BMDMs were treated with media (untreated), Wnt3a, IL-4 ,and Wnt3a+IL-4 for 16 hr. RNA was isolated as described above. Single-read RNA sequencing was performed using the Illumina HiSeq 2500. All samples had high quality, when checked with FastQC v0.11.7 (http://www.bioinformatics. babraham.ac.uk/projects/fastqc/). Illumina adapter sequences and poor-quality bases were trimmed using trimmomatic v0.36 (*Bolger et al., 2014*). Trimmed sequences were mapped to the mm10 mouse reference genome using STAR v2.6.0a (*Dobin et al., 2013*), indexed using samtools v1.9 (*Li et al., 2009*), then quantified for UCSC genes using HTSeq-count v0.11.1 (*Anders et al., 2015*). Comparative analysis between conditions was performed using DESeq2 v1.24.0 (*Love et al., 2014*) with default parameters. PCA was performed on the RNA (19,862 transcripts) dataset using PCA function in MATLAB (R2017a).

## Feature and violin plots

We summed the expression values of genes associated with the GO term GO:0016055 (Wnt signaling pathway), added the summed expression as a metadata feature using Seurat, and plotted the expression in UMAP using Seurat FeaturePlot. We also plotted the level of summed Wnt expression for each cell, summarized as a violin plot, and split by treatment group. Significance was assessed using the Wilcoxon rank-sum test within each cluster between groups.

## Heat map

For each of gene associated with GO:0016055, we calculated a log fold-change (logFC) value for resolution/progression for each cluster using the Seurat FindMarkers function. We then performed hierarchical clustering based on the gene logFC values and visualized them using pheatmap.

## Human plaque macrophage analysis

The sum and average of the logFC across macrophage clusters in the mouse dataset were analyzed for GO:0016055. The top 10 genes with the highest sum were the same as the top 10 genes with the highest average, and these genes were chosen for further analysis. LogFC was calculated for symptomatic versus asymptomatic patients. To determine the significance of the enrichment for negative logFC values (symptomatic/asymptomatic) in the subset of 10 Wnt responsive genes, we calculated logFC values (symptomatic/asymptomatic) for all genes in the dataset and ran a permutation test with 10,000 permutations. Specifically, for each of the permutations, we randomly selected 10 genes from the set of all genes and recorded the number of negative logFC values in that random selection. p-Values were calculated as the fraction of permutations that yielded seven or more negative logFC values. Human studies consent was obtained as described in *Fernandez et al., 2019*; *Alkhalil et al., 2018*.

## Western blotting

BMDM lysates were run in 12% SDS-polyacrylamide gels under denaturing conditions and blots were stained with anti-pY-STAT3 (9131;Cell Signaling Technology), anti-total-STAT3 (8768; Cell Signaling Technology), β-catenin (sc-7963; Santa Cruz), and anti-GAPDH antibodies (G8795; Sigma-Aldrich). Immunoreactive bands were visualized using the Odyssey imaging system (LI-COR Biosciences) by using horseradish peroxidase-conjugated secondary antibodies (ab205718; Abcam and 626520; Thermo Fisher Scientific) and chemiluminescence.

## Targeted liquid chromatography-tandem mass spectrometry

Media of BMDMs treated with Wnt3a was immediately added to two volumes of ice-cold methanol and stored at −80°C (*Dalli et al., 1730*). Prior to solid phase extraction, 500 pg of deuterated synthetic standards (d5-RvD2, d5-LXA$_4$, d4-LTB$_4$, and d4-PGE$_2$; Cayman Chemical) were added to the samples. Samples were centrifuged (3000 rpm for 10 min at 4°C) and supernatants collected, acidified to pH 3.5, and extracted using C18 cartridges (Biotage). Finally, lipid mediators were eluted from the column by addition of methyl formate. The solvent was then evaporated under N$_2$ gas.

Samples were resuspended in a 1:1 mixture of methanol and water and subjected to LC-MS/MS analysis using a Poroshell reverse-phase C18 column (100 mm × 4.6 mm × 2.7 µm; Agilent Technologies) equipped high-performance liquid chromatography system (Shimadzu) coupled to a QTrap 5500 mass-spectrometer (AB Sciex) operating in negative ionization mode. Scheduled multiple reaction monitoring (MRM) transitions coupled with information-dependent acquisition and enhanced product ion-scanning was used thereafter. Specific lipid mediators were identified by matching their retention time and at least six diagnostic MS/MS ions, compared with external standards (Cayman Chemical) analyzed in parallel. For quantification, the MRM peak was compared with synthetic standards. Extraction recovery was determined using the internal deuterated standards added to the samples prior to extraction.

## Statistical analysis

GraphPad Prism 7 (GraphPad Software) was used for statistical analysis. Data are expressed as mean ± SEM. With three or more groups, statistical analysis was performed using one-way ANOVA, with Dunnett's multiple comparisons testing and Gaussian distribution. Comparison of two parameters for two or more groups was performed using a two-way ANOVA, with Sidak's multiple comparison testing. Data involving comparisons between two groups were analyzed using two-tailed $t$-test. $p < 0.05$ was considered significant.

# Acknowledgements

We thank Drs. Thomas Wynn and Richard Locksley for providing mice used in this study. We also thank Mark Graham and Richard Lee (Ionis Pharmaceuticals) for graciously providing the ApoB ASO. SPH is supported by the NIH (K23HL135398). CG is supported by the American Heart Association (AHA, 20SFRN35210252) and the NIH (R03HL13528, K23HL111339, R21TR001739, UH2/3TR002067). DF is supported by NIH training grant 5T23HL007824. MS acknowledges the support of NIH grants HL106173 and GM095467. BES was supported by a National Research Service Award from the NIH (HL136044). This research was supported in part by the Intramural Research Program of the NIH to PL, as well as NIH grants AI130945, AI133977, HL084312, and U.S. Department of Defense (DoD) award W81XWH-16-1-0256. EAF acknowledges support from NIH grants HL084312 and DoD award W81XWH-16-1-0256. AW was supported by the AHA (18POST34080390) and the NIH (K99HL151963). KR was supported by NIH training grants T32GM007308 and T32AI100853, and NIH fellowship F30HL131183.

# Additional information

## Funding

| Funder | Grant reference number | Author |
| --- | --- | --- |
| National Heart, Lung, and Blood Institute | K23HL135398 | Sean P Heffron |
| National Heart, Lung, and Blood Institute | HL136044 | Brian E Sansbury |
| Division of Intramural Research, National Institute of Allergy and Infectious Diseases | AI130945 | P'ng Loke |
| Division of Intramural Research, National Institute of Allergy and Infectious Diseases | AI133977 | P'ng Loke |
| National Heart, Lung, and Blood Institute | HL084312 | P'ng Loke Edward A Fisher |
| U.S. Department of Defense | W81XWH-16-1-0256 | P'ng Loke Edward A Fisher |
| American Heart Association | 18POST34080390 | Ada Weinstock |
| Division of Intramural Research, National Institute of | T32AI100853 | Karishma Rahman |

Allergy and Infectious Diseases

| National Heart, Lung, and Blood Institute | F30HL131183 | Karishma Rahman |
| National Heart, Lung, and Blood Institute | K99HL151963 | Ada Weinstock |
| American Heart Association | 20SFRN35210252 | Chiara Giannarelli |
| National Heart, Lung, and Blood Institute | R03HL13528 | Chiara Giannarelli |
| National Heart, Lung, and Blood Institute | K23HL111339 | Chiara Giannarelli |
| National Heart, Lung, and Blood Institute | R21TR001739 | Chiara Giannarelli |
| National Heart, Lung, and Blood Institute | UH2/3TR002067 | Chiara Giannarelli |
| National Heart, Lung, and Blood Institute | 5T23HL007824 | Dawn Fernandez |
| National Heart, Lung, and Blood Institute | HL106173 | Matthew Spite |
| National Heart, Lung, and Blood Institute | GM095467 | Matthew Spite |
| NIH | T32GM007308 | Karishma Rahman |

The funders had no role in study design, data collection and interpretation, or the decision to submit the work for publication.

## Author contributions

Ada Weinstock, Conceptualization, Data curation, Formal analysis, Investigation, Methodology, Writing - original draft, Writing - review and editing; Karishma Rahman, Or Yaacov, Prashanthi Menon, Brian E Sansbury, Data curation, Investigation, Methodology; Hitoo Nishi, Data curation, Formal analysis; Cyrus A Nikain, Michela L Garabedian, Stephanie Pena, Gregory Marecki, Investigation; Naveed Akbar, Data curation, Formal analysis, Investigation; Sean P Heffron, Conceptualization, Methodology; Jianhua Liu, Investigation, Methodology; Dawn Fernandez, Emily J Brown, Stephen A Ramsey, Data curation, Formal analysis, Methodology; Kelly V Ruggles, Formal analysis, Methodology; Chiara Giannarelli, Conceptualization, Data curation, Supervision; Matthew Spite, Data curation, Formal analysis, Supervision, Methodology; Robin P Choudhury, Conceptualization, Supervision, Investigation, Methodology; P'ng Loke, Conceptualization, Data curation, Supervision, Writing - review and editing; Edward A Fisher, Conceptualization, Data curation, Supervision, Funding acquisition, Methodology, Writing - review and editing

## Author ORCIDs

Ada Weinstock https://orcid.org/0000-0003-3619-3388
Or Yaacov http://orcid.org/0000-0002-8496-2607
Kelly V Ruggles http://orcid.org/0000-0002-0152-0863
P'ng Loke http://orcid.org/0000-0002-6211-3292
Edward A Fisher https://orcid.org/0000-0001-9802-143X

## Ethics

Animal experimentation: This study was performed in strict accordance with the recommendations in the Guide for the Care and Use of Laboratory Animals of the National Institutes of Health. All of the animals were handled according to the protocol (number IA16-00494) approved by the Institutional Animal Care and Use Committee of the New York University School of Medicine.

## Decision letter and Author response

Decision letter https://doi.org/10.7554/eLife.67932.sa1

Author response https://doi.org/10.7554/eLife.67932.sa2

## Additional files

### Supplementary files

- Supplementary file 1. qPCR primer sequences.

- Transparent reporting form

### Data availability

The RNA sequencing data are deposited in GEO under accession number GSE168542.

The following dataset was generated:

| Author(s) | Year | Dataset title | Dataset URL | Database and Identifier |
| --- | --- | --- | --- | --- |
| Weinstock A, Ruggles KV, Brown EJ, Fisher EA | 2021 | Wnt signaling enhances macrophage responses to IL-4 and promotes resolution of atherosclerosis | https://www.ncbi.nlm.nih.gov/geo/query/acc.cgi?acc=GSE168542 | NCBI Gene Expression Omnibus, GSE168542 |

The following previously published datasets were used:

| Author(s) | Year | Dataset title | Dataset URL | Database and Identifier |
| --- | --- | --- | --- | --- |
| Fernandez DM, Rahman AH, Fernandez N, Chudnovskiy A, Amir ED, Amadori L, Khan NS, Wong CK, Shamailova R, Hill C, Wang Z, Remark R, Li JR, Pina C, Faries C, Awad AJ, Moss N, Bjorkegren JLM, Kim-Schulze S, Gnjatic S, Ma'ayan A, Mocco J, Faries P, Merad M, Giannarelli C | 2019 | Single-cell immune landscape of human atherosclerotic plaques | https://github.com/giannarelli-lab/Single-Cell-Immune-Profiling-of-Atherosclerotic-Plaques | Zenodo, 10.5281/zenodo.3361716 |
| Lin J, Nishi H, Poles J, Niu X, Brown EJ, Ramsey SA, Fisher EA, Loke P | 2019 | Single-cell analysis of fate-mapped macrophages reveals heterogeneity, including stem-like properties, during atherosclerosis progression and regression | https://www.ncbi.nlm.nih.gov/geo/query/acc.cgi?acc=GSE123587 | NCBI Gene Expression Omnibus, GSE123587 |
| Sharma M, Schlegel PM, Afonso MS, Brown EJ, Rahman K, Weinstock A, Sansbury BE, Corr EM, van Solingen C, Koelwyn GJ, Shanley LC, Beckett L, Peled D, Lafaille JJ, Spite M, Loke P, Fisher EA, Moore KJ | 2020 | Regulatory T cells license macrophage pro-resolving functions in atherosclerosis regression | https://www.ncbi.nlm.nih.gov/geo/query/acc.cgi?acc=GSE141038 | NCBI Gene Expression Omnibus, GSE141038 |
| Sanin DE, Matsushita M, Geltink RIK, Grzes KM, van Teijlingen Bakker N, Corrado M, Kabat AM, Buck MD, Qiu J, Lawless | 2019 | Mitochondrial Membrane Potential Regulates Nuclear Gene Expression in Macrophages Exposed to PGE2 (RNA-seq) | https://www.ncbi.nlm.nih.gov/geo/query/acc.cgi?acc=GSE119509 | NCBI Gene Expression Omnibus, GSE119509 |

SJ, Cameron AM,
Villa M, Baixauli F,
Patterson AE,
Hässler F, Curtis
JD, O'Neill CM,
O'Sullivan D, Wu D,
Mittler G, Huang
SC, Pearce EL,
Pearce EJ

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
