## [Decision Letter]

[Editors' note: this paper was reviewed by Review Commons.]

**Acceptance summary:**

This paper sought to investigate the roles of two interleukins, IL-4 and IL-13, in atherosclerosis aka hardened arteries. The authors note that global IL-4 and IL-13 deficiency impairs atherosclerosis resolution, but unexpectedly that the interleukin levels do not change during resolution of atherosclerosis. This led them to a discover a model whereby wnt3a signaling induces prostaglandin (PGE2) production and STAT3 activation. The combination of reduced lipid intake activating wnt3a/PGE2/STAT3 combined with STAT6 activation downstream of IL-4 signaling, led to a transcriptional synergy serving to promote atherosclerosis resolution. Overall, this manuscript helps define a combined pathway whereby inflammation and lipids together interact to help determine the fate of atherosclerosis.

---

## [Author Response]

We first want to thank the reviewers for their enthusiasm and support, as well as for insightful and comprehensive comments that we addressed to improve our manuscript.

Reviewer 1:Figure 1. I am missing the control experiment using Il4/13-proficient mice. To make the claim that IL4/13 are needed for significant atherosclerosis resolution that control (although in general published already) would be good to have in the same experimental set-up. Or is everything really identical to the Sharma et al. study (BL6 mice, PCSK9-AAV construct and administration conditions)? If so, this should be clearly stated and perhaps the experimental details re-iterated in Materials and methods.

The study design and mouse model are indeed identical to Sharma et al. We have extended the Materials and methods section to explain the model more thoroughly.

Figure 3. The cytokine analysis experiment in Figure 3A is not fully convincing. Chemokines such as MCP-1 (which BTW should be termed CCL2 to abide by current nomenclature) are well known to be endothelially deposited due to their high GAG binding property and to be prestored in vesicles. Can authors comment on how IL4 and IL-13 behave in this context? Can washout effects be excluded? Furthermore, could the almost undetectable levels of IL-13 be due to a washout effect? Along the same line: were the ELISA data confirmed by qPCR? If IL-13 levels were "underestimated", it may not have been fully justified to subsequently only focus on IL-4?

The name of the chemokine was altered in the figure and text to CCL2. Due to the experimental procedure, we do not believe that there was a washout of IL-13: plaques were excised and placed in a tube containing PBS and proteinase inhibitor. Plaques were then physically disrupted with a homogenizer and suspensions were measured directly, without any washing steps. In the same assay we also measured additional cytokines, which are not the focus of this manuscript, but are added in Author response image 1 for reviewer’s appreciation.

Additionally and in accordance with the reviewer’s suggestion, we examined the expression IL-13 in our previously published scRNAseq datasets of atherosclerosis resolution, and added these to Figure 3—figure supplement 1. The data demonstrate that IL-13 was only expressed in 1 cell in the entire dataset, further validating our observation of undetectable IL-13 levels.Furthermore, although less relevant in the context of plaque resolution, we believe our findings of enhancement of the Th2 cytokine response by Wnt extends to IL-13 as well. We examined the expression of related genes in macrophages treated with Wnt3a and IL-13 and observed a very similar increase in the expression to the response observed with IL-4 + Wnt3a treatment (n=9):

**Author response image 2. respfig2:** 

Figure 4. The human correlation data for Axin2 in Figure 4F is very intriguing and convincing, but also raises the questions whether other Wnt signature genes such a β-catenin showed a similar correlation?

As per the reviewer’s suggestion, we examined the association of plaque volume change with the expression of β catenin in the human study. Results show no association between the two parameters. With that said, β-catenin is mostly regulated at the protein level upon Wnt activation, whereas Axin2 is mostly transcriptionally regulated (via β-catenin protein transcriptional activation; Dzobo K et al. Journal of Integrative Biology, 2019), which likely accounts for these differences.

**Author response image 3. respfig3:** 

Figure 5. While the data show that the combination of Wnt3a and IL4 enhances the expression of Arg1, Ccl17, and Socs1, authors should refine the use of the term "synergistic". Synergism is actually only observed if one considers the addition of IL-4 to the Wnt3a-treated incubation. If one compares the IL-4 effect with that of the IL-4 + Wnt3a combination, the effect sees is "only" "additive".

We agree with the reviewer and refined these statements throughout the manuscript.

Also, the synergistic effect of IL-4 regarding activation by Wnt3a should also be shown on the level of (rapid) signal transduction, e.g. by looking at rapid signaling effects downstream of Wnt3a/FRZ/LRP (phosphorylation, protein level).

The molecular mechanism demonstrated in the manuscript is the activation of STAT3 (i.e., its phosphorylation), via the production of PGE2 by Wnt-stimulated macrophages. To investigate the rapidity of this signal transduction, we treated primary macrophages with Wnt3a or its conditioned media for 15, 30 and 60 minutes, which did not cause STAT3 phosphorylation by the indicated time points. Conditioned media from cells incubated with Wnt3a for 16 hours, however, did promote STAT3 phosphorylation within this short time frame as early as 15 minutes (Figure 7—figure supplement 1E). We are happy move these data to the main figures, if the reviewer thinks it will be more appropriate there.

Figure 7. Based on the data in Figure 7, authors insinuate the IL-6/STAT3 axis to synergize with IL-4 in skewing plaque macrophages towards a reparative/resolution phenotype. How do they reconcile the suggested role of IL-6 as a downstream target of IL-1b in promoting atherosclerotic inflammation (CANTOS)?

The reviewer brings up an interesting point, and the dichotomy of STAT3 as a factor either anti (in response to IL-10) or pro-inflammatory (in response to IL-6) is of interest to many. The experiment described in this manuscript, in which both IL-10 and IL-6 enhanced the response to IL-4 was a demonstration of the cooperation between STAT6 and STAT3 when both are activated. Our interpretation of the data is that the STAT3 activator in response to Wnt is PGE_2_, and not IL-6, as supported by the experiments in which PGE_2_ receptor binding was blocked (add any other PGE_2_-based strategy we used). It is likely that factors upstream or downstream of STAT3 may affect the selection of its transcriptional targets.

Figure 8. In the cartoon in Figure 8, authors imply a "paracrine" mechanism of Wnt3a-driven PGE2 production and PGE2-mediated STAT3 activation. Could this also happen in an "autocrine" manner? Please consider and discuss!

The reviewer is absolutely right, as we do not know if the effects are paracrine, autocrine or both. We, thus, revised our cartoon and added this point to the Discussion.

The Introduction is a bit "thin" and could be expanded.

The Introduction was expanded.

Cytokine names IL-4 and IL-13 should be written with "hyphen", i.e. "IL-4" not "IL4" as usually spelled in the field.

The names were revised to include a hyphen.

I would fully label the Il4/Il13-DKO as "Il4^-/-^ Il13^-/-^". The chosen abbreviation "Il4/Il13^-/-^" could create confusion as it doesn't clearly indicate that the Il4 gene is knocked out as well.

The abbreviation was revised, as suggested.

This sentence in results is confusing and seems disconnected: "Prior to transplant/lipid reduction, cells in the plaques are sufficient for IL4/13, while cells recruited post-transplant are deficient." Does it refer to an actual experiment? If so, were IL-4 and IL-13 measured?

This sentence was part of the explanation for the experiment. We removed the sentence.

Results section, it should read "Wnt3a protein and IL4" instead of “Wnt3a protein with IL4".

This was revised, as suggested.

Reviewer 2:I respectfully point out that Figure 1 is missing the WT controls so it is difficult to interpret the data. In general, I agree with the authors that the DKO are resistant to the regression model but whether that is because their plaques are very small or don't shrink cannot be formally deduced from the experimental layout. So, both wording and conclusions should be chosen more carefully.

Please find response to reviewer 1 (answer 1) in regards to the mouse model. We agree with the reviewer that detecting shrinkage of very small plaques would be a challenge. However, the plaques analyzed in Figure 1 are of average size of 319,293µm^2^ and we have previously demonstrated effective atherosclerosis resolution in plaques smaller than 200,000µm^2^ (Sharma et al., 2020 and Basu et al., 2018). Moreover, it is possible that the IL-4/13 DKO mice have smaller plaques than WT mice after Western Diet feeding, but given that plaque size is substantial, we think that the conclusion that these cytokines are needed for disease resolution holds. As suggested by the reviewer, we added a Discussion section for this potential caveat and thank both reviewers for raising this issue.

In the end, it remains unclear to me what the authors propose which cells signal the Wnt pathway where and when during atherogenesis and regression. I think the authors need a better model to grasp their hypotheses.

The focus of this manuscript is the cells *responding* to Wnt (macrophages), which become pro-reparative in response. We agree with the reviewer that the cells which produce Wnt and the exact time of response to Wnt is still unknown and is beyond the scope of this manuscript. Indeed, these questions are an active area of investigation in our group.

What is the plasma count of monocytes (more detailed by FACS, not just total leukocyte count) in the experiments? Do the altered numbers in the plaque reflect altered invasion or overall altered numbers, e.g. in the Wnt inhibitor experiment.

We added flow-cytometry data of Ly6c hi and Ly6c low monocytes to Figure 6—figure supplement 1F. The data show similar frequencies of these cells in the circulation, such that increased proportions of these cells is probably not the reason for the impaired resolution upon Wnt inhibition.

Some of the stats are stated correctly but I find it hard to believe that these numbers provide significance based on the data points. I would actually redo the math if I was reviewing the final version (see Figure 7F).

Thank you for the meticulous examination of the data. We repeated the analysis and received the same values as presented in the figure. In author response table 1 are the raw expression values for your additional review:

**Table resptable1:** 

		Control siRNA			STAT3 siRNA			
Biological replicate	Untreated	Wnt3a	IL4	Wnt+IL4	Untreated	Wnt3a	IL4	Wnt+IL4
1	0.000107	0.00025	0.017898	0.065526	0.000236	0.00018	0.00649	0.015214
2	7.54E-05	0.000137	0.004711	0.020549	4.2E-05	1.21E-05	0.001987	0.009788
3	0.000107	0.00025	0.017898	0.065526	0.000236	0.00018	0.00649	0.015214
4	0.000373	0.000343	0.000846	0.01827	0.000193	0.000239	0.000954	0.004027
5	9.18E-05	0	0.003173	0.009634	6.84E-05	0.000105	0.001183	0.001899

Performing 1-way ANOVA only of the STAT3 siRNA conditions still did not yield statistical significance, whereas a t-test of IL-4 versus Wnt+IL-4 resulted in a significant p-value of 0.02. With that said, we believe that a 2-way ANOVA is the most appropriate test for these data.

Green/red, red/blue on black for IMF/IHC (Figure 6F, 7G, is not ideal, I would use black/white color scale – enhances contrast for the readers and is inclusive.

Images were revised, as suggested by the reviewer.

The scheme depicting the experimental approach in the figures (1A, 2A, 3B, 6A etc.) can be more detailed.

We agree with the reviewer and added more details to the experimental approach figures.

The Introduction is very short compared to the rest. I would like to read more about what is known already in more detail.

The Introduction was expanded.